# Graph Persistence goes Spectral

**Mattie Ji**
University of Pennsylvania
mji13@sas.upenn.edu

**Amauri H. Souza**
Federal Institute of Ceará
amauriholanda@ifce.edu.br

**Vikas Garg**
Aalto University
YaiYai Ltd
vgarg@csail.mit.edu

## Abstract

Including intricate topological information (e.g., cycles) provably enhances the expressivity of message-passing graph neural networks (GNNs) beyond the Weisfeiler-Leman (WL) hierarchy. Consequently, Persistent Homology (PH) methods are increasingly employed for graph representation learning. In this context, recent works have proposed decorating classical PH diagrams with vertex and edge features for improved expressivity. However, these methods still fail to capture basic graph structural information. In this paper, we propose SpectRe — a new topological descriptor for graphs that integrates spectral information into PH diagrams. Notably, SpectRe is strictly more expressive than PH and spectral information on graphs alone. We also introduce notions of global and local stability to analyze existing descriptors and establish that SpectRe is locally stable. Finally, experiments on synthetic and real-world datasets demonstrate the effectiveness of SpectRe and its potential to enhance the capabilities of graph models in relevant learning tasks. Code is available at https://github.com/Aalto-QuML/SpectRe/.

## 1 Introduction

Relational data is ubiquitous in real-world applications, and can be elegantly abstracted with graphs. GNNs are state-of-the-art models for graph representation learning [10, 11, 28, 36, 51, 55, 62]. Almost all commonly employed GNNs can be cast as schemes where nodes repeatedly exchange messages with their neighbors [25]. Despite empirical success, GNNs are known to have some key limitations.

Notably, due to their strong local inductive bias, these GNNs and their higher-order counterparts are bounded in power by the WL hierarchy [40, 41, 43, 62]. Furthermore, they fail to compute important graph properties such as cycles and connectivity [15, 24]. Topological descriptors such as those based on PH can provide such information not just for GNNs but also the so-called topological neural networks (TNNs) that generalize message-passing to higher-dimensional topological domains, enabling more nuanced representations than the standard GNNs [6–8, 22, 26, 27, 46, 47, 56].

Specifically, PH employs *filtrations* (or filter functions) that can track the evolution of key topological information; e.g., when a new component starts or the time interval during which a component survives (until two components merge, or indefinitely). This persistence information is typically encoded as (birth, death) pairs, or more generally tuples with additional entries, in a persistence diagram. The topological features derived from these persistence diagrams can be integrated into GNNs and TNNs to enhance their expressivity and boost their empirical performance [56]. PH is thus increasingly being utilized in (graph) machine learning [1, 9, 13, 14, 29–31, 49, 54, 63, 64, 66, 67].

Understandably, there is a growing interest in designing more expressive PH descriptors for graphs [2]. Recently, Immonen et al. [33] analyzed the representational ability of *color-based* PH schemes, providing a complete characterization of the power of $0$-dimensional PH methods that employ vertex-level or edge-level filtrations using graph-theoretic notions. They also introduced RePHINE as a strictly more powerful descriptor than these methods. However, it turns out that RePHINE is still unable to

Table 1: Overview of our theoretical results.

| **Expressive Power of Filtration Methods (Section 3)** | |
| --- | --- |
| Construction of SpectRe Diagrams | Definition 3.1 |
| SpectRe is isomorphism invariant | Theorem 3.2 |
| SpectRe $\succ$ RePHINE and SpectRe $\succ$ Laplacian Spectrum | Theorem 3.3 |
| **Stability of RePHINE and SpectRe Diagrams (Section 4):** | |
| Construction of a suitable metric $d_B^R$ on RePHINE | Definition 4.2 |
| Construction of a suitable metric $d_B^{\text{Spec } R}$ on SpectRe | Definition 4.3 |
| RePHINE is globally stable under $d_B^R$ | Theorem 4.4 |
| SpectRe is locally stable under $d_B^{\text{Spec } R}$ | Theorem 4.5 |
| Estimate on the Extent of Instability for $d_B^{\text{Spec } R}$ | Theorem 4.7 |
| Verifying $d_B^R$ and $d_B^{\text{Spec } R}$ are metrics | Proposition A.2 |

separate some simple non-isomorphic graphs: e.g., on monochromatic graphs, RePHINE recovers the same information as vanilla PH. We, therefore, seek to design a more expressive PH descriptor here.

Our key idea is to enhance the persistence tuples of RePHINE with the evolving spectral information inherent in the subgraphs resulting from the filtration. Spectral information has been previously found useful in different learning tasks over graphs [4, 5, 32, 37, 38, 57, 60], which motivates our investigations into extracting spectral signatures from the graph Laplacian.

Laplacian appears in several flavors: unlike graph Laplacian (the 0-th combinatorial Laplacian), rows in the 1-st combinatorial Laplacian correspond to the edges of the graph (as opposed to vertices), and higher-dimensional persistent versions [59] have also been proposed for both. Since we are interested in graph-based filtrations, simply tracking the graph Laplacian of filtered subgraphs provides all the additional expressivity that the higher-order generalizations of the Laplacian can offer. Guided by this key insight, we introduce a new topological scheme called SpectRe that amalgamates RePHINE and graph Laplacians to be strictly more expressive than both these methods.

Furthermore, we unravel the theoretical merits of SpectRe. Specifically, stability is a key desideratum of PH descriptors [16]. Stability of PH is typically assessed via a *bottleneck distance*, which provides a suitable metric on the space of persistence diagrams (obtained from different filtration functions). The bottleneck distance guarantees a minimum separation between the filtered spaces, which makes persistence diagrams an effective tool in several applications such as shape classification and retrieval [3]. Stability results are known for standard persistence diagrams in [16, 39, 58]. However, since PH metrics for the persistence diagrams in edge-level filtrations are agnostic of the node color, even defining a suitable metric to quantify stability is challenging for methods such as RePHINE (that strictly generalizes PH with node colors). We first fill this gap with a novel generalization for the bottleneck distance that forms a metric on the space of persistence diagram of RePHINE, and establish that RePHINE is globally stable under this metric. Interestingly, we proceed to show that SpectRe is not globally stable but only locally stable (which suffices for practical applications) under a similar metric, outside of a measure zero subset of all possible filtrations. Moreover, we quantify an explicit upper bound on how unstable SpectRe can be when "crossing over a locus of instability" in terms of the complexity of the input graph.

To validate our theoretical analysis and show the effectiveness of SpectRe diagrams, we conduct experiments using two sets of synthetic datasets for assessing the expressive power of graph models (13 datasets in total), and multiple real datasets. Overall, these results demonstrate the higher expressivity of SpectRe and illustrate its potential for boosting the capabilities of graph neural networks on graph classification. We summarize our theoretical results in Table 1, and relegate all the proofs to Appendix.

## 2 Preliminaries

Unless mentioned otherwise, we will consider graphs $G = (V, E, c, X)$ with a finite vertex set $V$, edges $E \subseteq V \times V$, and a vertex-coloring function $c : V \to X$, where $X$ is a finite set denoting the

space of available colors or features. All graphs are simple unless mentioned otherwise. Two graphs $G = (V, E, c, X)$ and $G' = (V', E', c', X')$ are **isomorphic** if there is a bijection $h : V \to V'$ of the vertices such that (1) the two coloring functions are related by $c = c' \circ h$ and (2) the edge $(v, w)$ is in $E$ if and only if $(h(v), h(w))$ is in $E'$.

We remark the first condition ensures that isomorphic graphs should share the same coloring set. For example, the graph $K_3$ with all vertices colored "red" will not be isomorphic to the graph $K_3$ with all vertices colored "blue", as it fails the first condition. For the rest of this work, we will assume that **all graphs that appear have the coloring set** $X$ (we can always, without loss of generality, take $X$ to be the union of their coloring sets). In this work, we are interested in graph features that change over time (i.e. persistent descriptors) as opposed to static features. Our notion of time will be a color filtration of a graph defined as follows.

**Definition 2.1** (Coloring Filtrations). On a color set $X$, we choose a pair of functions $(f_v : X \to \mathbb{R}, f_e : X \times X \to \mathbb{R}_{>0})$ where $f_e$ is symmetric (i.e. $f_e(a, b) = f_e(b, a)$ for all $a, b \in X$). On a graph $G$ with a vertex color set $X$, the pair $(f_v, f_e)$ induces the following pair of functions $(F_v : V \cup E \to \mathbb{R}, F_e : V \cup E \to \mathbb{R}_{\geq 0})$.

1. For all $v \in V(G)$, $F_v(v) := f_v(c(v))$. For all $e \in E(G)$ with vertices $v_1, v_2$, $F_v(e) = \max\{F_v(v_1), F_v(v_2)\}$. Intuitively, we are assigning the edge $e$ with the color $c(\arg\max_{v_i} F_v(v_i))$ (the vertex color with a higher value under $f_v$).
2. For all $v \in V(G)$, $F_e(v) = 0$. For all $e \in e(G)$ with vertices $v_1, v_2$, $F_e(e) := f_e(c(v_1), c(v_2))$. Intuitively, we are assigning the edge $e$ with the color $(c(v_1), c(v_2))$.

For each $t \in \mathbb{R}$, we write $G_t^{f_v} := F_v^{-1}((-\infty, t])$ and $G_t^{f_e} := F_e^{-1}((-\infty, t])$.

Note we chose the notation "$G_t^{f_v}$ and $G_t^{f_e}$" as opposed to "$G_t^{F_v}$ and $G_t^{F_e}$" to emphasize that the function $G \mapsto (\{G_t^{f_v}\}_{t \in \mathbb{R}}, \{G_t^{f_e}\}_{\mathbb{R}})$ is well-defined for any graph $G$ with the common coloring set $X$. The lists $\{G_t^{f_v}\}_{t \in \mathbb{R}}$ and $\{G_t^{f_e}\}_{t \in \mathbb{R}}$ define a **vertex filtration** of $G$ by $F_v$ and an **edge filtration** of $G$ by $F_e$ respectively. It is clear that $G_t^{f_v}$ can only change when $t$ crosses a critical value in $\{f_v(c) : c \in X\}$, and $G_t^{f_e}$ can only change when $t$ crosses a critical value in $\{f_e(c_1, c_2) : (c_1, c_2) \in X \times X\}$. Hence, we can reduce both filtrations to finite filtrations at those critical values.

## 2.1 Persistent Homology on Graphs

The core idea of persistent homology is to track how topological features evolve throughout a filtration, accounting for their appearance/disappearance. In particular, we say a vertex $v$ (i.e. 0-dimensional persistence information) is born when it first appears in a given filtration. When we merge two connected components represented by two vertices $v$ and $w$, we use a decision rule to kill off one of the vertices and mark the remaining vertex to represent the new connected component. Similarly, a cycle (i.e. 1-dimensional persistence information) is born when it appears in a filtration, and it will never die. For a vertex $v$ or an edge $e$, we mark its **persistence pair** as the tuple $(b, d)$, where $b$ and $d$ indicate its birth and death time respectively (here $d = \infty$ if the feature never dies).

For color-based vertex and edge filtrations, there is a canonical way to calculate the persistence pairs of a graph for a given filtration. We refer the reader to Appendix A of Immonen et al. [33] for a precise introduction. Following the terminology in Immonen et al. [33], we say a 0-th dimensional persistence pair $(b, d)$ is a **real hole** if $d = \infty$, is an **almost hole** if $b \neq d < \infty$, and is a **trivial hole** if $b = d$. Note that edge-based filtrations do not have any trivial holes.

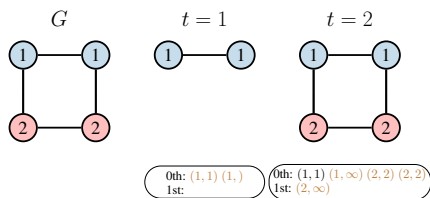

Figure 1: Vertex-level PH: filtration and diagram induced by $f_v$. Here, we have that $f_v(\text{blue}) = 1$ and $f_v(\text{red}) = 2$.

**Definition 2.2.** Let $f = (f_v, f_e)$ be on the coloring set $X$, as in Definition 2.1. The **persistent homology (PH) diagram** of a graph $G$ is a collection $\text{PH}(G, f)$ composed of two lists $\text{PH}(G, f)^0, \text{PH}(G, f)^1$ where $\text{PH}(G, f)^0$ are all persistence pairs in the vertex filtration $\{G_t^{f_v}\}_{t \in \mathbb{R}}$ and $\text{PH}(G, f)^1$ are all pairs in the edge filtration $\{G_t^{f_e}\}_{t \in \mathbb{R}}$.

Figure 1 depicts a vertex filtration along with its 0-th dim PH diagram $\text{PH}(G, f)^0$. We also provide in Figure 2 the persistence pairs of $\text{PH}(G, f)^1$ for the same graph.

## 2.2 RePHINE

Despite the growing popularity of PH in graph representation learning, PH alone does not fully capture local color information. To overcome this limitation, Immonen et al. [33] introduced RePHINE as a generalization of PH.

**Definition 2.3.** Let $f = (f_v, f_e)$ be on $X$. The **RePHINE diagram** of a graph $G$ is a multi-set $\mathrm{RePHINE}(G, f) = \mathrm{RePHINE}(G, f)^0 \sqcup \mathrm{RePHINE}(G, f)^1$ of cardinality $|V(G)| + \beta_G^1$ where:

- 0-th dimensional component: $\mathrm{RePHINE}(G, f)^0$ consists of tuples of the form $(b(v), d(v), \alpha(v), \gamma(v))$ for each vertex $v \in V(G)$. Here, $b(v)$ and $d(v)$ are the birth and death times of $v$ under the edge filtration $\{G_t^{f_e}\}_{t \in \mathbb{R}}$, $\alpha(v) = f_v(c(v))$ and $\gamma(v) = \min_{\omega \in N(v)} f_e(c(v), c(w))$, where $N(v)$ denotes the neighboring vertices of $v$.

  The decision rule for which vertex to kill off is as follows - an almost hole $(b, d)$ corresponds to the merging of two connected components with vertex representatives $v_1$ and $v_2$. We kill off the vertex with a greater value under $\alpha$. If there is a tie, we kill off the vertex with a lower value under $\gamma$. If there is a further tie, Theorem 4 of Immonen et al. [33] shows that the resulting diagram $\mathrm{RePHINE}(G, f)^0$ is independent of which vertex we kill off here.

- 1-st dimensional component: $\mathrm{RePHINE}(G, f)^1$ consists of tuples of the form $(1, d(e), 0, 0)$ for each $e$ in the first persistence diagram. $d(e)$ indicates the birth time of a cycle in the same filtration. In the definition of Immonen et al. [33], the birth of a cycle corresponds to what is called the death of a "missing hole". This is why we use $d$ to indicate the birth time instead.

Theorem 5 of Immonen et al. [33] asserts that $\mathrm{RePHINE}$ diagrams are strictly more expressive than PH diagrams.

# 3 Spectrum-informed Persistence Diagrams

We now introduce a novel descriptor, called $\mathrm{SpectRe}$, that incorporates spectral information. We also demonstrate its isomorphism invariance and analyze its expressive power.

## 3.1 Laplacian Spectrum and SpectRe

Homology captures harmonic information (in the sense that the kernel of the graph Laplacian corresponds to the 0-th homology), but there is some non-harmonic information we also want to account for. One option is to account for colors, as we have done with RePHINE. Another option is to augment RePHINE with spectral information. Building on this idea, we propose a new descriptor below.

**Definition 3.1.** Let $f = (f_v, f_e)$ be on the coloring set $X$. The **spectral RePHINE diagram** (SpectRe, in short) of a graph $G$ is a multi-set $\mathrm{SpectRe}(G, f) = \mathrm{SpectRe}(G, f)^0 \sqcup \mathrm{SpectRe}(G, f)^1$ of cardinality $|V(G)| + \beta_G^1$ where:

- 0-th dimensional component: $\mathrm{SpectRe}(G, f)^0$ consists of tuples of the form $(b(v), d(v), \alpha(v), \gamma(v), \rho(v))$ for each vertex $v \in V(G)$. Here, $b, d, \alpha, \gamma$ are the same as Definition 2.3, and $\rho(v)$ is the list of non-zero eigenvalues of the graph Laplacian of the connected component $v$ is in when it dies at time $d(v)$.

- 1-st dimensional component: $\mathrm{SpectRe}(G, f)^1$ consists of tuples of the form $(1, d(e), 0, 0, \rho(e))$ for each $e$ in the first persistence diagram. Here, $d(e)$ indicates the birth time of a cycle given by the edge $e$. $\rho(e)$ denotes the non-zero eigenvalues the graph Laplacian of the connected component $e$ is in when it is born at time $d(e)$.

For completeness, we also define the **Laplacian spectrum (LS) diagram** of a graph $G$ as the projection of $\mathrm{SpectRe}(G)$ to its $b$-component, $d$-component, and $\rho$-component. Figure 2 shows an example of computing $\mathrm{SpectRe}, \mathrm{RePHINE}$, and LS on the same graph and filtration.

## 3.2 Expressive Power

Here, we compare the expressivity of SpectRe to both RePHINE and Laplacian spectrum, showing that SpectRe is strictly more expressive than the other two. We also discuss an alternative, more

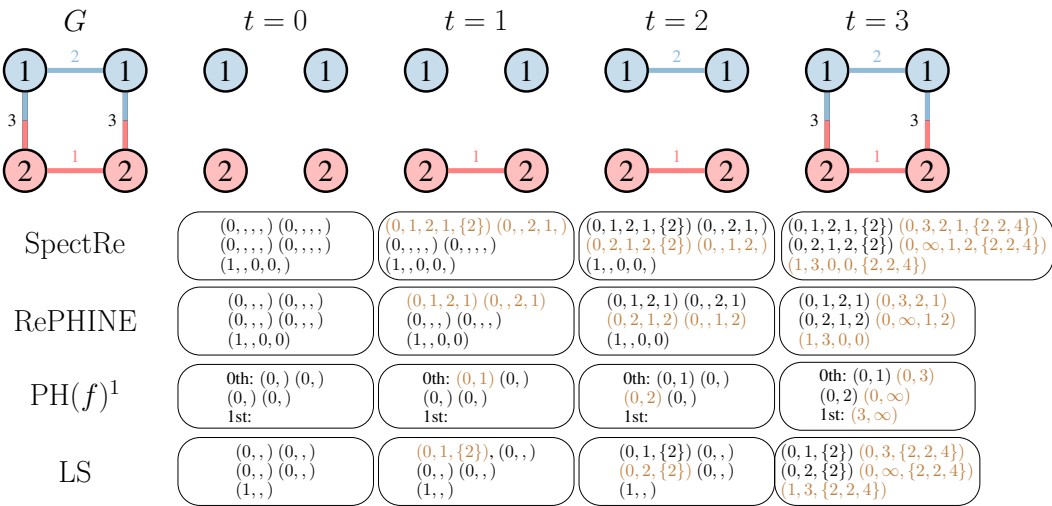

Figure 2: Example computing SpectRe, RePHINE, (edge-level) PH, and LS on a graph $G$ with $f_v(\text{blue}) = 1$, $f_v(\text{red}) = 2$ and $f_e(\text{red}) = 1$, $f_e(\text{blue}) = 2$, $f_e(\text{red-blue}) = 3$. The graph $G$ has an edge filtration by $f_e$.

elaborate definition of SpectRe we initially considered using the ideas of a "persistent Laplacian" in Wang et al. [59]. The upshot is that we show this approach is as expressive as our current definition.

To be precise on what we mean by *expressivity*, let $X, Y$ be two graph isomorphism invariants. We say $X$ has **at least the same expressivity** as $Y$ (denoted $X \succeq Y$) if for all non-isomorphic graphs $G$ and $H$ that $Y$ can tell apart, $X$ can also tell them apart. We say $X$ is **strictly more expressive** than $Y$ (denoted $X \succ Y$) if, in addition, there exist two non-isomorphic graphs $G$ and $H$ that $Y$ cannot tell apart but $X$ can. We say $X$ and $Y$ have the **same expressive power** (denoted $X = Y$) if $X \succeq Y$ and $Y \succeq X$. We say $X$ and $Y$ are **incomparable** if there are two non-isomorphic graphs $G$ and $H$ that $X$ can tell apart but $Y$ cannot, and vice versa.

Before analyzing SpectRe's expressivity, we must first verify its invariance to graph isomorphism.

**Theorem 3.2.** *Suppose $G$ and $H$ are isomorphic graphs with the same color set $X$. Let $f = (f_v, f_e)$ be any filtration functions on $X$, then $\mathrm{SpectRe}(G, f)$ is equal to $\mathrm{SpectRe}(H, f)$.*

Now we state the expressivity comparisons of SpectRe, RePHINE, and the Laplacian spectrum (LS).

**Theorem 3.3.** SpectRe *is strictly more expressive than both* RePHINE *and* LS*, which are incomparable to each other. Furthermore, the counterexamples are illustrated in Figure 3.*

The intuition behind why RePHINE cannot differentiate the examples in Figure 3 is that it has the same amount of expressive power as counting the number of connected components and independent cycles (i.e. the harmonic components of the Laplacian) on a monochromatic graph (see Lemma C.1). Introducing spectral information (i.e. the non-harmonic components of the Laplacian) can give lens to a wider scope to distinguish these graphs.

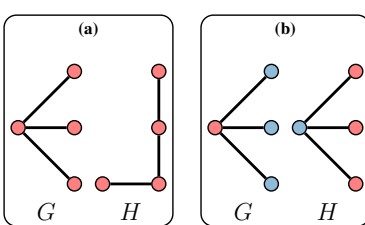

Figure 3: (a) Graphs that SpectRe and LS can separate but not RePHINE. (b) Graphs that SpectRe and RePHINE can separate but not LS.

It is well-known that the kernel of the graph Laplacian of $G$ has the same dimension as the number of connected components (i.e. 0-th Betti number) of $G$. There is a generalization of graph Laplacians to "combinatorial Laplacians" (whose kernels yield higher Betti numbers) and even more generally "a persistent version of Laplacians" [59] (whose kernels yield the persistent Betti numbers). One might ask whether incorporating their spectrum into RePHINE would result in greater expressivity than SpectRe.

**Proposition 3.4.** *Consider an alternative topological descriptor on $G = (V, E, c, X)$, $f = (f_v, f_e)$ given by $\Phi(f) := \{(b(v), d(v), \alpha(v), \gamma(v), \rho'(v)\}_{v \in V} \sqcup \{(b(e), d(e), \alpha(e), \gamma(v), \rho'(e)\}_{e \in \{\textit{ind. cycles}\}}$, where the first four components are the same as* RePHINE *(Definition 2.3), $\rho'(v)$ returns the spectrum of the persistent Laplacians in [59] on the connected component $v$ dies in, and $\rho'(e)$ returns the spectrum of the persistent Laplacians of the connected component $e$ is born in.*

*If we remember which vertex and edge is assigned to which tuple in the computation, $\Phi$ has the same expressive power as* SpectRe.

In other words, augmenting RePHINE with the spectrum of persistent Laplacians is just as expressive as including the spectrum of the graph Laplacian. Note that the assumption for remembering the tuple assignment is always satisfied in practice, as computing SpectRe, RePHINE, or PH-based methods in general would go through such assignments (see Algorithm 1 and 2 of [33]). It then suffices for us to focus on graph Laplacians hereafter.

We provide a self-contained proof of Proposition 3.4 in Appendix B for completeness. Note that the comparison in Proposition 3.4 can be derived from a fact noted in Example 2.3 of [45] and Section 6 of [19]. We remark that it is unclear how Proposition 3.4 would follow without assuming the underlined condition that we remember the tuple assignment, which is a situation that [19, 45] did not encounter since their design of descriptors is different from ours.

## 4  Stability of RePHINE and SpectRe

Let $f = (f_v : X \to \mathbb{R}, f_e : X \times X \to \mathbb{R}_{>0})$ and $g = (g_v : X \to \mathbb{R}, g_e : X \times X \to \mathbb{R}_{>0})$ be two pairs of functions on $X$. For a graph $G$ with the coloring set $X$, we would ideally like a way to measure how much the diagrams we constructed in Section 2 differ in SpectRe of $f$ and $g$. One way to measure this is to impose a suitable metric on the space of diagrams and obtain a stable bound. For PH diagrams on $G$, this suitable metric is called the **bottleneck distance** and has been classically shown to be bounded by $||f_v - g_v||_\infty + ||f_e - g_e||_\infty$ [16]. In this section, we will discuss a generalization of this result to RePHINE and SpectRe diagrams.

Let us examine RePHINE first. One naive proposal for a suitable "metric" on RePHINE diagrams would be to restrict only to the first two components ($b$ and $d$) of the multi-set and use the classical bottleneck distance for persistence diagrams. However, this approach ignores any information from the $\alpha$ and $\gamma$ components and will fail the non-degeneracy axioms for a metric. Thus, we need to modify the metric on RePHINE diagrams to take into account of its $\alpha$ and $\gamma$ components.

For ease of notation, we introduce the following definition to simplify our constructions.

**Notation 4.1.** Let $A$ and $B$ be two finite multi-subsets of the same cardinality in a metric space $(X, d_X)$, we write $\text{Bott}(A, B, d_X) := \inf_{\pi \in \text{bijections A to B}} \max_{p \in A} d_X(p, \pi(p))$. Informally, $\text{Bott}(A, B, d_X)$ is the infimum of distances for which there is a bijection between $A$ and $B$ in $X$.

Note that when $X = \mathbb{R}^2$ and $d_X$ is the $\ell_\infty$-norm, we recover the definition of bottleneck distance for PH diagrams. Now we will define a metric for RePHINE diagrams.

**Definition 4.2.** Let $\text{RePHINE}(G, f)$ and $\text{RePHINE}(G, g)$ be the two associated RePHINE diagrams for $G$ respectively. We define the **bottleneck distance** as $d_B^R(\text{RePHINE}(G, f), \text{RePHINE}(G, g)) := d_B^{R,0}(\text{RePHINE}(G, f)^0, \text{RePHINE}(G, g)^0) + d_B^{R,1}(\text{RePHINE}(G, f)^1, \text{RePHINE}(G, g)^1)$.

Here, $d_B^{R,0}(\bullet, \bullet)$ and $d_B^{R,1}(\bullet, \bullet)$ are both given by $\text{Bott}(\bullet, \bullet, d)$, where $d$ is defined on $\mathbb{R}^4$ as

$$d((b_0, d_0, \alpha_0, \gamma_0), (b_1, d_1, \alpha_1, \gamma_1)) = \max\{|b_1 - b_0|, |d_1 - d_0|\} + |\alpha_1 - \alpha_0| + |\gamma_1 - \gamma_0|.$$

Similarly, we define a metric for SpectRe diagrams.

**Definition 4.3.** We define the **bottleneck distance** between SpectRe diagrams as

$$d_B^{\text{Spec } R}(\text{SpectRe}(G, f), \text{SpectRe}(G, g)) := d_B^{\text{Spec } R,0}(\text{SpectRe}(G, f)^0, \text{SpectRe}(G, g)^0)$$
$$+ d_B^{\text{Spec } R,1}(\text{SpectRe}(G, f)^1, \text{SpectRe}(G, g)^1)$$

Here, $d_B^{\text{Spec } R,0}(\bullet, \bullet)$ and $d_B^{\text{Spec } R,1}(\bullet, \bullet)$ are both given by $\text{Bott}(\bullet, \bullet, d')$, where $d'$ is defined on $\mathbb{R}^5$ as

$$d'((b_0, d_0, \alpha_0, \gamma_0, \rho_0), (b_1, d_1, \alpha_1, \gamma_1, \rho_1)) = d((b_0, d_0, \alpha_0, \gamma_0), (b_1, d_1, \alpha_1, \gamma_1)) + d^{Spec}(\rho_0, \rho_1).$$

Here, $d$ is given in Definition 4.2 and $d^{Spec}$ is given by embedding $\rho_0$ and $\rho_1$ as sorted lists (followed by zeroes) into $\ell^1(\mathbb{N})$ and taking their $\ell^1$-distance in $\ell^1(\mathbb{N})$.

We verify in Appendix A.2 that Definition 4.2 and Definition 4.3 are indeed metrics. We also prove that the bottleneck distances of two RePHINE diagrams may be explicitly bounded in terms of the $\ell^\infty$ norms of the input functions, and hence RePHINE diagrams are stable in the following sense.

> **Theorem 4.4.** *For any choice of $f = (f_v, f_e)$ and $g = (g_v, g_e)$ as before, we have the inequality*
>
> $$d_B^R(\mathrm{RePHINE}(G, f), \mathrm{RePHINE}(G, g)) \leq 3||f_e - g_e||_\infty + ||f_v - g_v||_\infty.$$

RePHINE diagrams are regarded as *globally stable* in the sense that no matter what $f$ and $g$ we choose, their respective RePHINE diagrams are bounded by a suitable norm on $f$ and $g$. SpectRe diagrams, in contrast, only satisfy a local form of stability. We make precise what *local* means by introducing a suitable topology on the possible space of filtration functions.

After fixing a canonical ordering on $X$ and $X \times X/ \sim$ separately, we may view $f_v$ (resp. $f_e$) as an element of $\mathbb{R}^{n_v}$ (resp. $(\mathbb{R}_{>0})^{n_e}$). Furthermore, if $f_v$ (resp. $f_e$) is injective, it may viewed as an element in $\mathrm{Conf}_{n_v}(\mathbb{R})$ (resp. $\mathrm{Conf}_{n_e}(\mathbb{R}_{>0})$), where $\mathrm{Conf}_{n_v}(\mathbb{R})$ (resp. $\mathrm{Conf}_{n_e}(\mathbb{R}_{>0})$) is the subspace of $\mathbb{R}^{n_v}$ (resp. $(\mathbb{R}_{>0})^{n_e}$) composing of points whose coordinates have no repeated entries. From here we obtain the following theorem.

> **Theorem 4.5.** *If $f_v$ and $f_e$ are injective, then $f = (f_v, f_e)$ is locally stable on $\mathrm{Conf}_{n_v}(\mathbb{R}) \times \mathrm{Conf}_{n_e}(\mathbb{R}_{>0})$ under $d_B^{\mathrm{Spec}\,R}$. That is, over a graph $G$ with the coloring set $X$, we have:*
>
> $$d_B^{\mathrm{Spec}\,R}(\mathrm{SpectRe}(G, f), \mathrm{SpectRe}(G, g)) \leq 3||f_e - g_e||_\infty + ||f_v - g_v||_\infty$$
>
> *for all $g = (g_v, g_e)$ sufficiently close to $f$ in $\mathrm{Conf}_{n_v}(\mathbb{R}) \times \mathrm{Conf}_{n_e}(\mathbb{R}_{>0})$. Furthermore, the same bound holds without imposing the local stability condition in $f_v$.*

Local stability suffices in practical applications. Note that the injectivity assumption is necessary for $f_e$, and $\mathrm{SpectRe}$ is not globally stable in general. We illustrate this in the following example.

**Example 4.6.** Let $G$ be the path graph on $4$ vertices colored in the order red, blue, blue, red. Let $f_v = g_v$ be any functions. Let $g_e$ be constant with value 1, and $f_e$ be given by $f_e(\mathrm{red}, \mathrm{blue}) = 1$, $f_e(\mathrm{blue}, \mathrm{blue}) = 1 - \epsilon$ for $\epsilon > 0$ small, and $f_e(\mathrm{red}, \mathrm{red})$ any value. $\mathrm{SpectRe}(G, f)^0$ has 1 tuple representing 1 blue vertex that dies at time $t = 1 - \epsilon$ whose $\rho$-parameter is $\{2\}$. In contrast, every tuple of $\mathrm{SpectRe}(G, g)^0$ has its $\rho$-parameter equal to the list $L = \{2, 2 \pm \sqrt{2}\}$. No matter how small $\epsilon > 0$ is, the distance on their $\mathrm{SpectRe}$ diagrams is bounded below by

$$d_B^{\mathrm{Spec}\,R}(\mathrm{SpectRe}(G, f), \mathrm{SpectRe}(G, g)) \geq d^{Spec}(\{2\}, L) > 0.$$

Thus, $\mathrm{SpectRe}$ could fail to be locally stable without the injectivity assumption.

We also note that $\mathrm{SpectRe}$ is not globally stable, even with the injectivity assumption. A counter-example can be found by changing $g_e$ in the same example to the function given by $g_e(\mathrm{red}, \mathrm{red}) = f_e(\mathrm{red}, \mathrm{red})$, $g_e(\mathrm{red}, \mathrm{blue}) = 1$, and $g_e(\mathrm{blue}, \mathrm{blue}) = 1 + \epsilon$ for $\epsilon > 0$. The intuitive reason is that to change $g_e(\mathrm{blue}, \mathrm{blue}) = 1 + \epsilon$ to $f_e(\mathrm{blue}, \mathrm{blue}) = 1 - \epsilon$, $g_e$ has to become non-injective when its value at $(\mathrm{blue}, \mathrm{blue})$ crosses 1.

We can, however, obtain a bound to how unstable the situation becomes when injectivity is violated.

**Theorem 4.7.** *Let $G$ be a graph and let $f_e$ be an injective edge filtration function with values $a_1 < ... < a_n$ with $a_i = f(e_i)$ for each edge color type $e_i$. Let $g_e$ be a perturbation of $f_e$ such that $g_e(e_j) = f_e(e_j)$ for all $j \neq i$, and $g_e(e_i) = g_e(e_{i+1}) = a_{i+1}$ (ie. $g_e$ merges the i-th and i+1-th value together). Let $f_v, g_v$ be vertex filtration functions, then the distance between the **0-dimensional components** of $\mathrm{SpectRe}(G, f_v, f_e)$ and $\mathrm{SpectRe}(G, g_v, g_e)$ is bounded by*

$$||g_v - f_v||_\infty + 2(a_{i+1} - a_i) + 2 \max_{(H_1, H_2) \in Y} |E(H_2)| - |E(H_1)|,$$

*where $Y$ is the collection of pairs $(H_1, H_2)$ where $H_1$ is the connected graph that a vertex v died at time $a_i$ for $f_e$ is contained in, $H_2$ is the connected graph the same vertex $v$ died at time $a_{i+1}$ for $g_e$ is in, and $H_1 \subset H_2$.*

*The distance between the **1-dimensional components** of $\mathrm{SpectRe}(G, f_v, f_e)$ and $\mathrm{SpectRe}(G, g_v, g_e)$ is bounded by*

$$(a_{i+1} - a_i) + 2 \max_{(H_1, H_2) \in Z} |E(H_2)| - |E(H_1)|,$$

*where $Z$ is the collection of pairs $(H_1, H_2)$ where $H_1$ is the connected graph that a cycle-creating edge $e$ born at $a_i$ for $f_e$ is contained in, $H_2$ is the connected graph for the same edge $e$ at time $a_{i+1}$ for $g_e$, and $H_1 \subset H_2$.*

We conclude this section by remarking that analyzing the stability of $\mathrm{RePHINE}$ and $\mathrm{SpectRe}$ is more challenging than one might expect. For RePHINE, the $\alpha$ values can be interpreted as the birth time of the vertices, but there was no clear interpretation of what the $\gamma$ values are in terms of the birth/death time of a simplex, rendering a direct application of the stability of PH fruitless. For SpectRe, the filtration method was not globally stable, which came as a surprise to us. To circumvent this finding, we instead came up with a new description of the topology of the filtration functions and showed it is locally stable under this topology. For both RePHINE and SpectRe, it was also unclear how the setting of Bottleneck stability would change since the original metric on persistence pairs in [16] was given by the $\ell_\infty$-norms, but we adopted various other norms in this work.

## 5 Integration with GNNs

Like most topological descriptors for graphs, SpectRe can be seamlessly integrated into GNNs. Following Immonen et al. [33], we use GNN representations at each layer as inputs to filtration functions and obtain a vectorized representation of the entire diagram by encoding its tuples with DeepSets [65]. However, unlike existing methods, we must also encode the spectrum (the $\rho$ component) associated with each tuple. To this end, we use an additional DeepSet model dedicated to processing these spectral features. Consequently, at each GNN layer, we obtain a new SpectRe diagram which we then vectorize (as described above). These representations are then integrated into the GNN — for example, by concatenating them with the graph-level GNN's embedding — just before the final classification head.

Formally, let $\mathrm{SpectRe}(G^k, f^k)^0$ be the 0-th dimensional SpectRe diagram obtained at the $k$-th GNN layer. Since each element of the diagram is associated with a node $v$, we compute the topological embedding $r_k^0$ of $\mathrm{SpectRe}(G^k, f^k)^0$ as

$$\tilde{r}_{v,k}^0 = \phi_k^1 \left( \sum_{p \in \rho(v)} \psi_k^1(p) \right) \qquad r_k^0 = \phi_k^2 \left( \sum_{v \in V} \psi_k^2 \left( b(v), d(v), \alpha(v), \gamma(v), \tilde{r}_{v,k}^0 \right) \right), \qquad (1)$$

where $\phi_k^1, \phi_k^2, \psi_k^1, \psi_k^2$ are feedforward neural networks. We follow an analogous procedure to obtain 1-dimension topological embeddings at each layer.

It is worth noting that there are alternative ways to integrate topological embeddings with GNNs. For instance, instead of computing a diagram-level representation, Horn et al. [31] incorporates vertex-level representations, $\tilde{r}_{v,k}^0$, by adding them to the node embeddings of the GNN at each layer.

Table 2: Accuracy in distinguishing all pairs of minimal Cayley graphs with $n$ nodes (c-$n$) and graphs in the BREC datasets. We use degree as vertex filter ($f_v$) and Forman–Ricci curvature as edge filter ($f_e$). SpectRe separates all Cayley graphs and is the best performing method on BREC datasets. Note that some outputs are slightly different from earlier versions of the paper due to a minor bug in the code (see Appendix E for further details).

| | Cayley Graphs | | | | | | | | BREC Datasets | | | | | |
| Methods | c-12 | c-16 | c-20 | c-24 | c-32 | c-36 | c-60 | c-63 | B (60) | R (50) | E (100) | C (100) | D (20) | All (400) |
| --- | --- | --- | --- | --- | --- | --- | --- | --- | --- | --- | --- | --- | --- | --- |
| PH$^0$ | 0.67 | 0.83 | 0.61 | 0.65 | 0.76 | 0.69 | 0.69 | 0.49 | 0.03 | 0.00 | 0.07 | 0.03 | 0.00 | 0.03 |
| PH$^1$ | 0.95 | 0.83 | 0.61 | 0.86 | 0.76 | 0.84 | 0.78 | 0.73 | 0.98 | 0.94 | 0.55 | 0.03 | 0.00 | 0.41 |
| RePHINE | 0.95 | 0.83 | 0.61 | 0.86 | 0.76 | 0.84 | 0.78 | 0.73 | 0.98 | 0.94 | 0.55 | 0.03 | 0.00 | 0.41 |
| SpectRe | **1.00** | **1.00** | **1.00** | **1.00** | **1.00** | **1.00** | **1.00** | **1.00** | **1.00** | **1.00** | **1.00** | **0.04** | **0.05** | **0.54** |

# 6 Experiments

To assess the effectiveness of SpectRe from an empirical perspective, we consider two sets of experiments. The first consists of isomorphism tests on synthetic benchmarks designed to evaluate the expressive power of graph models. The second explores the combination of topological descriptors and GNNs in real-world tasks. Implementation details are provided in Appendix E. In addition, our code is publicly available at https://github.com/Aalto-QuML/SpectRe/.

## 6.1 Synthetic data

Following Ballester and Rieck [2], we consider datasets containing minimal Cayley graphs with varying numbers of nodes [18] and the BREC benchmark [61] - details are given in the Appendix. We compare four topological descriptors (PH$^0$, PH$^1$, RePHINE, and SpectRe) obtained using fixed filtration functions. Specifically, we use node degrees and augmented Forman–Ricci curvatures [50] as vertex- and edge-level filtration functions, respectively — i.e., $f_v(u) = |N(u)|$ and $f_e(u, w) = 4 - |N(u)| - |N(w)| + 3|N(u) \cap N(w)|$. Note that only RePHINE and SpectRe leverage both functions as PH$^0$ applies $f_e(u, w) = \max\{f_v(u), f_v(w)\}$ and PH$^1$ uses $f_v(u) = 0$ for all $u$.

Table 2 presents the accuracy results for the Cayley and BREC datasets. Here, accuracy represents the fraction of pairs of distinct graphs for which the multisets of persistence tuples differ. In all datasets, the performance of PH$^0$ is bounded by that of PH$^1$, while PH$^1$ performs on par with RePHINE. Interestingly, for most datasets, these descriptors distinguish exactly the same graphs. Notably, SpectRe can separate all pairs of minimal Cayley graphs. Overall, PH$^0$ struggles to distinguish graphs across all BREC datasets, indicating that degree information is not informative in these

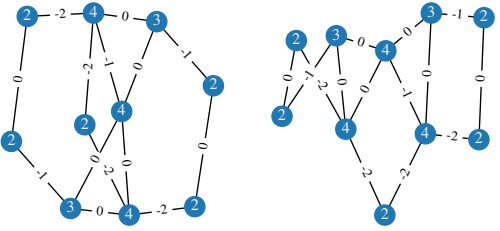

Figure 4: Example of a pair of graphs from the dataset Extension(100) that RePHINE cannot distinguish but SpectRe can. Node and edge labels denote filtration values.

cases. Except for the (regular) Distance dataset, these results confirm the high expressivity of SpectRe on exploiting complex graph structures, even under simple degree-based filtrations. We note that the BREC datasets 'strongly connected' and '4-vertex condition' were omitted from the table since no descriptor can separate any of their pairs of graphs. For completeness, Figure 4 illustrates a pair of graphs that RePHINE does not separate but SpectRe does.

## 6.2 Real data

To illustrate the potential of SpectRe on boosting the power of GNNs, we consider nine datasets for graph-level predictive tasks: MUTAG, PTC-MM, PTC-MR, PTC-FR, NCI1, NCI109, IMDB-B, ZINC, MOLHIV [42]. Again, we wish to compare SpectRe against PH$^0$ (vertex filtrations) and RePHINE. Thus, we consider the same GNN architecture for all diagrams: graph convolutional network (GCN) [36]. We provide additional results using graph Transformers in the Appendix. Importantly, all diagrams are vectorized exactly the same way using DeepSets. We report the mean and standard deviation of accuracy (MAE for ZINC, and AUROC for MOLHIV) over three independent runs.

Table 3: Predictive performance on graph classification/regression. We denote the highest mean accuracy (lowest MAE for ZINC) in bold. For most datasets, SpectRe is the best performing model.

| Method | MUTAG | PTC-MM | PTC-MR | PTC-FR | NCI1 | NCI109 | IMDB-B | ZINC | MOLHIV |
|--------|-------|--------|--------|--------|------|--------|--------|------|--------|
| GCN | 63.2±4.7 | 56.9±3.3 | 58.1±1.6 | 67.6±1.6 | 74.33±2.58 | 73.49±0.86 | 69.00±1.41 | 0.87±0.01 | 71.73±1.05 |
| $PH^0$ | 82.5±6.0 | 59.8±7.4 | 55.2±3.3 | 65.7±9.7 | 77.37±2.06 | 76.15±2.57 | 71.00±0.00 | 0.53±0.01 | 71.31±3.00 |
| RePHINE | 87.7±3.0 | 57.8±1.6 | 58.1±11.9 | 68.5±6.4 | 80.66±1.55 | 76.51±1.03 | 75.00±2.83 | **0.47**±0.01 | 76.03±0.48 |
| SpectRe | **91.2**±3.0 | **61.7**±2.9 | **59.1**±5.9 | **69.4**±2.8 | **80.90**±0.17 | **76.63**±0.86 | **76.00**±0.00 | 0.48±0.02 | **76.33**±0.57 |

To enable SpectRe for large datasets (e.g., NCI1, ZINC and MOLHIV), we employed the power method for computing the largest eigenvalue if $n > 9$ and used the full spectrum otherwise. In addition, we applied scheduling: we only computed the eigendecomposition at 33% of the filtration steps.

Table 3 shows the results on real-world data. SpectRe achieves the highest mean accuracies in 8 out of 9 cases, with a significant margin on the MUTAG and PTC-MM datasets. In most cases, the second best performing model is RePHINE. Overall, these results support the idea that GNNs can benefit from PH-based descriptors on real-world graph learning tasks.

**Additional experiments.** For completeness, Appendix F includes further experiments using Graph Transformers [48], along with ablation studies and an empirical assessment of our stability bounds.

**Limitations.** Like most highly expressive graph models, SpectRe's power comes at the cost of increased computational complexity. The overhead of computing eigendecomposition across graph filtrations is non-negligible and can become a bottleneck for large-scale real-world datasets. Specifically, on graphs, 0- and 1-dimensional persistence diagrams can be computed in $O(m \log m)$ time using disjoint sets, where $m$ is the number of edges. Since SpectRe requires eigendecomposition, in the worst case, the cost is $O(n^5)$ which comes from a fully connected graph with the eigendecomposition being applied to $\Theta(n^2)$ filtration steps. In the best case (considering connected graphs), the whole graph is revealed at the same time and the cost is $O(n^3)$. Fortunately, there are faster algorithms for computing (or approximating) partial spectral information. For instance, to run experiments on large datasets, one can employ LOBPCG or the power method, which reduces the time complexity to $O(kn^4)$ in the worst case, where $k$ is the number of iterations (often much smaller than $n$). If we further consider scheduling, i.e., only applying decomposition to a fixed number of filtration steps, we can further reduce the worst-case complexity to $O(kn^2)$.

In this work, we focused on color-based filtrations of graphs. However, exploring alternative constructions (e.g., biparameter or Vietoris–Rips filtrations) would be an interesting research direction. Also, while SpectRe incorporates spectral information with respect to edge-based filtrations, it would also be worthwhile to investigate how they interact with vertex-based filtrations. We discuss these directions in greater detail in Appendix D.

# 7 Conclusion

We augmented PH-based descriptors on GNNs with the Laplacian spectrum, focusing on expressivity, stability, and experiments. For expressivity, we amalgamated spectral features with RePHINE to craft a strictly more expressive scheme SpectRe than both RePHINE and the Laplacian spectrum. For stability, we constructed a notion of bottleneck distance on RePHINE and SpectRe. We showed that the former is globally stable, and the latter is locally stable. Building on our theoretical foundations, we proposed an integration of SpectRe with GNNs and "vectorized" the spectral component using DeepSets. We also benchmarked SpectRE against other TDs in this work on both synthetic and real data, showing empirical gains with our new approach.

# Broader Impact

While we do not anticipate immediate negative societal impacts from our work, we believe it can serve as a catalyst for the development of principled learning methods that effectively integrate topological information into graph representation learning. By enhancing the expressivity of graph-based models, such approaches hold promise for advancing a wide range of applications — including molecular modeling, drug discovery, social network analysis, recommender systems, and material design — where capturing multiscale structural and topological dependencies is essential.

## Acknowledgments and Disclosure of Funding

This research was conducted while the first author was participating during the 2024 Aalto Science Institute international summer research programme at Aalto University. We are grateful to the anonymous program chair, area chair and the reviewers for their constructive feedback and service. VG acknowledges Saab-WASP (grant 411025), Academy of Finland (grant 342077), and the Jane and Aatos Erkko Foundation (grant 7001703) for their support. AS acknowledges the Conselho Nacional de Desenvolvimento Científico e Tecnológico (CNPq) (312068/2025-5). AS would also like to thank Jorge Franco for his assistance with C++ routines. We also acknowledge the computational resources provided under the Aalto Science-IT Project by Computer Science IT. MJ would like to thank Sabína Gulčíková for her amazing help at Finland in Summer 2024. MJ would also like to thank Cheng Chen, Yifan Guo, Aidan Hennessey, Yaojie Hu, Yongxi Lin, Yuhan Liu, Semir Mujevic, Jiayuan Sheng, Chenglu Wang, Anna Wei, Jinghui Yang, Jingxin Zhang, an anonymous friend, and more for their incredible support in the second half of 2024.

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

# A  Proofs

## A.1  Proofs for Section 3

In this section, we will prove Theorem 3.2 and Theorem 3.3, and we discuss the proof of Theorem 3.4 separately in Appendix B.

*Proof of Theorem 3.2.* Let us check that SpectRe is a graph isomorphism invariant. We will first show this for $\mathrm{SpectRe}(\bullet, f)^0$. Here we follow the suggestions laid out in Theorem 4 of Immonen et al. [33] and decompose it in two steps.

1. From Theorem 4 of Immonen et al. [33], we know that the original RePHINE with $(b, d, \alpha, \gamma)$ is an isomorphism invariant.

2. We also want to show that the map $G \mapsto \{\rho(C(v, d(v)))\}_{v \in V(G)}$ is an isomorphism invariant. Here, $\rho$ produces the spectrum of the connected component $v$ is in at its time $d(v)$. Since $\rho$ itself does not depend on choice, it suffices for us to show the following map is an isomorphism invariant:

$$G \mapsto \{C(v, d(v))\}_{v \in V(G)},$$

   where $C(v, d)$ is the component $v$ is in at time $d$. The ambiguity comes in, depending on our choice, a vertex may very well die at different times.

   Now from the proof of Theorem 4 of Immonen et al. [33], we already know that the multi-set of death times is an isomorphism invariant. Now a vertex death can only occur during a merging of old connected components $T_1, ..., T_n$ (with representatives $v_1, ..., v_n$) to a component $C$. Now under the RePHINE scheme, there is a specific procedure to choose which vertex. However, we see that any choice of the vertex does not affect the connected component that will be produced after merging. Thus, we will always be adding a constant $n - 1$ copies of $C$ to the function we are producing.

   For the real holes, we know from the proof of Theorem 4 of Immonen et al. [33] that, it does not matter how each of the remaining vertices is matched to the real holes, since the rest of the vertices are associated in an invariant way. Hence, the production of graph Laplacians for the real holes will not be affected. Finally, the description above also shows that we can concatenate the pair $(b, d, \alpha, \gamma)$ with $(\rho)$ in a consistent way.

3. Hence, we see that $\mathrm{SpectRe}(\bullet, f)^0$ is an isomorphism invariant.

Now for $\mathrm{SpectRe}(\bullet, f)^1$, the argument follows similarly as above. It suffices for us to check that the list $\{C(e, b(e))\}_{e \in \mathrm{cycle}}$ is consistent. We observe that the birth of a cycle can only happen when an edge occurs that goes from a connected component $C$ back to itself. The only possible ambiguity is that, at the birth time, multiple edges are spawned at the same time, and an edge may or may not create a cycle based on the order it is added to the graph. However, regardless of the order, the resulting connected component that the edge belongs in will be the same. Thus, the list $\{C(e, b(e))\}_{e \in \mathrm{cycle}}$ will be consistent. $\qquad\square$

*Proof of Theorem 3.3.* Clearly SpectRe has at least the same expressivity to either RePHINE or LS. This is because RePHINE is the first four coordinates of SpectRe, and LS is the first, second, and fifth coordinates of SpectRe. Thus, it suffices for us to show that RePHINE and LS are incomparable to each other.

Consider the graphs $G$ and $H$ in Figure 3(a). From the remarks below Theorem 5 of Immonen et al. [33], we know that RePHINE cannot differentiate $G$ and $H$. However, the data about the real holes of the respective SpectRe diagrams of $G$ and $H$ will be different. This is because the eigenvalues of the graph Laplacian $\Delta_0(G)$ are $\{0, 1, 1, 4\}$ and the eigenvalues of the graph Laplacian $\Delta_0(H)$ are $\{0, 2, 2 - \sqrt{2}, 2 + \sqrt{2}\}$.

On the other hand, consider the graphs $G$ and $H$ in Figure 3(b). We note that all edges in $G$ and $H$ would be labeled the color "red-blue". Thus, the $\rho$-component of the LS diagrams for $G$ and

$H$ would be the same, being the non-zero eigenvalues of the star graph on 4 vertices. The set of death-times would also be the same, being $\{f_e(\text{red-blue}), f_e(\text{red-blue}), f_e(\text{red-blue}), \infty\}$. Thus, LS would not be able to differentiate $G$ and $H$. However, RePHINE would if we choose $f_v$ such that $f_v(\text{red}) \neq f_v(\text{blue})$, just based on comparing the $\alpha$-values. $\qquad\square$

## A.2  Proofs for Section 4

For ease of notation, we will omit the parameter $G$ in the RePHINE diagram in this section since we will only be discussing functions on the same graph. We will first verify that our definition of a metric $d_B^R$ in Definition 4.2 is actually a metric. To do this, we first remind the reader that, under Notation 4.1, the definition of $d_B^R$ can be elaborated as the following definition.

**Definition A.1.** Let $\text{RePHINE}(G, f)$ and $\text{RePHINE}(G, g)$ be the two associated RePHINE diagrams for $G$ respectively. We define the **bottleneck distance** as $d_B^R(\text{RePHINE}(G, f), \text{RePHINE}(G, g)) := d_B^{R,0}(\text{RePHINE}(G, f)^0, \text{RePHINE}(G, g)^0) + d_B(\text{RePHINE}(G, f)^1, \text{RePHINE}(G, g)^1)$.

For the 0-th dimensional component, $d_B^{R,0}$ is defined as,

$$d_B^{R,0}(\text{RePHINE}(G, f)^0, \text{RePHINE}(G, g)^0) := \inf_{\pi \in \text{bijections}} \max_{p \in \text{RePHINE}(G,f)^0} d(p, \pi(p)),$$

where $d$ is defined as $d((b_0, d_0, \alpha_0, \gamma_0), (b_1, d_1, \alpha_1, \gamma_1)) = \max\{|b_1 - b_0|, |d_1 - d_0|\} + |\alpha_1 - \alpha_0| + |\gamma_1 - \gamma_0|$, and $\pi$ ranges over all bijections $\text{RePHINE}(G, f)^0 \to \text{RePHINE}(G, g)^0$. For the 1-st dimensional component, $d_B$ is the usual bottleneck distance on 1-dimensional persistence pairs.

**Proposition A.2.** $d_B^R$ and $d_B^{\text{Spec}R}$ are metrics.

*Proof.* We will first check this for $d_B^R$. Since the usual bottleneck distance is a metric, it suffices for us to check that $d_B^{R,0}$ is a metric.

**1. Non-degeneracy:** Clearly the function is non-negative, and choosing $\pi$ to be the identity bijection shows that $d_B^{R,0}(\text{RePHINE}(f)^0, \text{RePHINE}(f)^0) = 0$. For any pairs $\text{RePHINE}(f)^0 \neq \text{RePHINE}(g)^0$, the term $\max_{p \in \text{RePHINE}(f)^0} d(p, \pi(p))$ will be greater than 0 for any choice of bijection $\pi$. Since there are only finitely many possible bijections $\pi$, the infimum $d_B^{R,0}(\text{RePHINE}(f)^0, \text{RePHINE}(g)^0)$ will be greater than 0.

**2. Symmetry:** Any bijection $\pi : \text{RePHINE}(f)^0 \to \text{RePHINE}(g)^0$ corresponds exactly to a bijection $\pi^{-1} : \text{RePHINE}(g)^0 \to \text{RePHINE}(f)^0$. Hence, the definition of $d_B^{R,0}$ is symmetric.

**3. Triangle Inequality:** Let $\text{RePHINE}(f)^0, \text{RePHINE}(g)^0, \text{RePHINE}(h)^0$ be the vertex components of RePHINE diagrams on $G$. Suppose $\sigma_1 : \text{RePHINE}(f)^0 \to \text{RePHINE}(h)^0$ is a bijection that achieves the infimum labeled in the definition of $d_B^{R,0}$. In other words,

$$d_B^{R,0}(\text{RePHINE}(f)^0, \text{RePHINE}(h)^0) = \max_{p \in \text{RePHINE}(f)^0} d(p, \sigma_1(p)).$$

Suppose $\tau_1, \tau_2$ are bijections from $\text{RePHINE}(f)^0 \to \text{RePHINE}(g)^0$ and $\text{RePHINE}(g)^0 \to \text{RePHINE}(h)^0$ respectively. We then have that,

$$\max_{p \in \text{RePHINE}(f)^0} d(p, \sigma_1(p)) \leq \max_{p \in \text{RePHINE}(f)^0} d(p, \tau_1(p)) + d(\tau_1(p), \tau_2(\tau_1(p)))$$

$$\text{Triangle Inequality for } d$$

$$\leq \max_{p \in \text{RePHINE}(f)^0} d(p, \tau_1(p)) + \max_{q \in \text{RePHINE}(g)^0} d(q, \tau_2(q)).$$

For the sake of paragraph space, we write $A = d_B^{R,0}(\text{RePHINE}(f)^0, \text{RePHINE}(h)^0)$. Taking infimum over all possible $\tau_1$ and over $\tau_2$ gives us that

$$A = d_B^{R,0}(\text{RePHINE}(f)^0, \text{RePHINE}(h)^0)$$

$$\leq \inf_{\tau_1, \tau_2} \max_{p \in \text{RePHINE}(f)^0} d(p, \tau_1(p)) + \max_{q \in \text{RePHINE}(g)^0} d(q, \tau_2(q))$$

$$\leq \inf_{\tau_1 \in \text{bijection}} \max_{p \in \text{RePHINE}(f)^0} d(p, \tau_1(p)) + \inf_{\tau_2 \in \text{bijection}} \max_{q \in \text{RePHINE}(g)^0} d(q, \tau_2(q))$$

$$\leq d_B^{R,0}(\text{RePHINE}(f)^0, \text{RePHINE}(g)^0) + d_B^{R,0}(\text{RePHINE}(g)^0, \text{RePHINE}(h)^0).$$

This shows that $d_B^{R,0}$ satisfies the triangle inequality.

For $d_B^{\mathrm{Spec}R}$, since we showed $d$ from Definition 4.2 is a metric, it suffices for us to check in this proposition that $d^{\mathrm{Spec}}$ is a metric. To reiterate the definition of $d^{\mathrm{Spec}}$, given a list $L$ of non-zero eigenvalues with length $n$, we define an embedding $\phi(L) \in \ell^1(\mathbb{N})$ where $\phi(L)$ where the first $n$-elements in the sequence are $L$ sorted in ascending order and the rest are zeroes. This embedding is clearly injective on the lists of non-zero eigenvalues. For two lists $\rho_0, \rho_1$, we define

$$d^{Spec}(\rho_0, \rho_1) = ||\phi(\rho_0) - \phi(\rho_1)||_1.$$

The fact that $d^{Spec}$ is a metric now follows from the fact that $\ell^1(\mathbb{N})$ is a metric space under its $\ell^1$-distance and $\phi$ is injective. $\qquad\square$

To prove Theorem 4.4, we first define a technical construction as follows.

**Definition A.3.** Given a graph $G$ and functions $(f_v, f_e)$ on $X$. This induces functions $(F_v, F_e)$ as defined in Definition 2.1. From here, we construct a pairing between $\mathrm{RePHINE}(f)^0$ and $(v, e) \in V(G) \times \{0\} \cup E(G)$ as follows.

1. For every almost hole $(0, d)$ that occurs in the edge filtration by $F_e$, this corresponds to the merging of two connected components represented by vertices $v_i$ and $v_j$.

2. We assign to $(0, d)$ the vertex that has greater value under $\alpha$. If there is a tie, we assign the vertex that has the lower value under $\gamma$. If there is a further tie, we will be flexible in how we assign them in the proof of the stability of the RePHINE diagram.

3. The occurrence of an almost hole $(0, d)$ is caused by an edge $e$ whose value under $f_e$ is $d$ that merges two connected components. We assign this edge to $(0, d)$. If there are multiple such edges, we will be flexible in how we assign them in the proof of the stability of the RePHINE diagram.

4. For the real holes, we assign them with the vertices left. The edge takes an uninformative value (i.e. $0$).

Note that for any vertex $v$ that dies at finite time $d$, its associated edge $e = (v_1, v_2)$ satisfies $f_e(c(v_1), c(v_2)) = d(v)$.

We will now state and prove two propositions that will directly imply Theorem 4.4. Our proof is inspired by the methods presented in Skraba and Turner [52].

*Remark* A.4. To expand on the original statement above: our proof relies on the technical construction in Definition A.3, which is inspired by vertex-edge pairing in Skraba and Turner [52] (which are called pivots) and proceeds with a linear interpolation technique as in Skraba and Turner [52]. Vertex-edge pairing has also appeared in Skraba and Turner [53] and Cohen-Steiner et al. [17] for stability results. While we focused on stability analysis here, the concept of vertex-edge pair is standard in TDA; e.g., it has appeared in numerous algorithmic designs in Dey and Wang [20].

Our definition of a pairing in Definition A.3 is different from the pivots in Skraba and Turner [52]. A pairing here is a pivot in Skraba and Turner [52] with respect to $f_e$, but not every pivot with respect to $f_e$ is a pairing. This is because there are some obstructions to conclude Theorem 4.4 directly from bottleneck stability, such as [16, 52]: (1) RePHINE has 4 parameters, while the stability results with traditional PH deals with only 2 parameters. (2) Our analysis here has to pay attention to a pair of filtrations $(f_v, f_e)$ compared to another pair $(g_v, g_e)$, as opposed to just comparing between individual filtration functions. (3) For the pair $(f_v, f_e)$, it is unclear how to interpret the $\gamma$ parameter in terms of a birth/death time of a simplex in $(f_v, f_e)$. For an arbitrary vertex $v \in G$, its $\gamma(v)$ need not be its birth time or its death time in $f_v/f_e$. It is possible to view $\gamma$ as the birth-time arising in another filtration $f_v'$, but we did not pursue this approach as our metrics are defined with respect to the pair $(f_v, f_e)$.

Extending the stability analysis from PH to RePHINE is non-trivial. In particular, we must resort to a RePHINE version of the vertex-edge pairing (Definition A.3) that respects both filtrations $(f_v, f_e)$. For our purpose, we require a pairing that respects both vertex and edge filtrations. This can be seen in Proposition A.5 below, whose proof uses an assignment with respect to $f_e$ to analyze changes to

$f_v$. The subtlety also occurs in Proposition A.7 that analyzes changes in $f_e$, since the proof entails a careful analysis on how the $\gamma$-parameter (which is not present in traditional PH) behaves.

**Proposition A.5.** *Suppose $f_e = g_e = h$ for some edge coloring function $h$, then*

$$d_B^{R,0}(\text{RePHINE}(f_v, h)^0, \text{RePHINE}(g_v, h)^0) \leq ||f_v - g_v||_\infty.$$

*Proof.* Let $h_v^t(x) = (1-t)f_v(x) + tg_v(x)$. Also let $H_v^t : G \to \mathbb{R}$ be the induced function of $h_v^t$ on $G$, in the sense of Definition 2.1. We can divide $[0,1]$ into finite intervals $[t_0, t_1], [t_1, t_2], ..., [t_n, t_{n+1}]$, where $t_0 = 0, t_{n+1} = 1, t_0 < t_1 < ... < t_{n+1}$, such that for all $t \in [t_i, t_{i+1}]$ and all simplicies $x, y \in G$, either

$$H_v^t(x) - H_v^t(y) \leq 0 \text{ or } \geq 0 \quad (\dagger).$$

To be clear on the wording, this means that we cannot find $s, s' \in [t_i, t_{i+1}]$ such that $H_v^s(x) > H_v^s(y)$ but $H_v^{s'}(x) < H_v^{s'}(y)$.

For all $s_1, s_2 \in [t_i, t_{i+1}]$, we claim that we can use Definition A.3 to produce the same list of pairs $(v, e)$ (with some flexible adjustments at endpoints if needed).

Since $f_e = g_e$, the list of death times and order of edges that appear do not change, what could change is which vertex to kill off at the time stamp. Let us now order the finite death times (i.e. those corresponding to almost holes), accounting for multiplicity, as $d_1 \leq d_2 \leq ... \leq d_n < \infty$. Now we observe that

1. At $d_1$, $\text{RePHINE}(h_v^{s_1}, h)$ and $\text{RePHINE}(h_v^{s_2}, h)$ will be merging the same two connected components with vertex representatives $v$ and $w$. For the RePHINE diagram at $s_1$ (resp. $s_2$), we choose which vertex to kill off based on which vertex has a higher value under $H_v^{s_1}$ (resp. $H_v^{s_2}$). By ($\dagger$), we will be killing off the same vertex. If there happens to be a tie of $\alpha$ values, we will still kill off the same vertex in the comparison of $\gamma$ values since $f_e = g_e$. Finally, if there is a tie of $\gamma$ values, we make the flexible choice to kill off the same vertex.

   Since $f_e = g_e$, the edge associated to this vertex can be chosen to be the same. Hence, $\text{RePHINE}(h_v^{s_1}, h)$ and $\text{RePHINE}(h_v^{s_2}, h)$ will produce the same pair $(v, e)$ at time $d_1$.

2. Suppose that up to the $i$-th death, both RePHINE diagrams are producing the same pairs and merging the same components. For the $i+1$-death, the RePHINE diagrams at both $s_1$ and $s_2$ will be merging the same two components $v'$ and $w'$. The same argument as the case for $d_1$ shows that they will produce the same pair of vertex and edge.

3. After we go through all finite death times, both RePHINE diagrams will have the same list of vertices that are not killed off, which are then matched to real holes.

This proves the claim above. From triangle inequality, we know that $d_B^{R,0}(\text{RePHINE}(f_v, h)^0, \text{RePHINE}(g_v, h)^0)$ is bounded by the term

$$\sum_{i=0}^{n} d_B^{R,0}(\text{RePHINE}(h_v^{t_i}, h)^0, \text{RePHINE}(h_v^{t_{i+1}}, h)^0).$$

For each summand on the right, we assign a bijection from $\text{RePHINE}(h_v^{t_i}, h)^0$ to $\text{RePHINE}(h_v^{t_{i+1}}, h)^0$ as follows - using our previous claim, we send $(0, d, \alpha, \gamma) \in \text{RePHINE}(h_v^{t_i}, h)^0$ to the pair in $\text{RePHINE}(h_v^{t_{i+1}}, h)^0$ that correspond to the same $(v, e)$. For the sake of paragraph space, we write $A_i = d_B^{R,0}(\text{RePHINE}(h_v^{t_i}, h)^0, \text{RePHINE}(h_v^{t_{i+1}}, h)^0)$ and

see that

$$
\begin{aligned}
A_i &= d_B^{R,0}(\text{RePHINE}(h_v^{t_i}, h)^0, \text{RePHINE}(h_v^{t_{i+1}}, h)^0) \\
&\leq \max_{(v,e)} |d_{t_{i+1}}(v) - d_{t_i}(v)| + |\alpha_{t_{i+1}}(v) - \alpha_{t_i}(v)| + |\gamma_{t_{i+1}}(v) - \gamma_{t_i}(v)| \\
&= \max_{(v,e)} 0 + |\alpha_{t_{i+1}}(v) - \alpha_{t_i}(v)| + 0 \qquad\qquad\qquad\qquad\text{Since } f_e = g_e \\
&= \max_v |\alpha_{t_{i+1}}(v) - \alpha_{t_i}(v)| \\
&= \max_w |h_v^{t_{i+1}}(c(w)) - h_v^{t_i}(c(w))| \\
&= \max_w |(1 - t_{i+1})f_v(c(w)) + t_{i+1}g_v(c(w)) - (1 - t_i)f_v(c(w)) - t_i g_v(c(w))| \\
&= \max_w |(t_i - t_{i+1})f_v(c(w)) + (t_{i+1} - t_i)g_v(c(w))| \\
&= \max_w (t_{i+1} - t_i)|f_v(c(w)) - g_v(c(w))| \\
&\leq (t_{i+1} - t_i)\|f_v - g_v\|_\infty.
\end{aligned}
$$

Hence, we have that

$$
\begin{aligned}
d_B^{R,0}(\text{RePHINE}(f_v, h)^0, \text{RePHINE}(g_v, h)^0) &\leq \sum_{i=0}^n A_i \\
&\leq \sum_{i=0}^n (t_{i+1} - t_i)\|f_v - g_v\|_\infty \\
&= \|f_v - g_v\|_\infty.
\end{aligned}
$$

$\square$

*Remark* A.6. In the proof of the previous proposition, we claimed that we can assign the same vertex-edge pair to each death time for both filtration at $t_i$ and $t_{i+1}$. This may seem contradictory at first, as this seems to suggest that, by connecting the endpoints of the interval, RePHINE would assign the same vertices on $G$ as real holes regardless of the choice of functions. However, in our proof, the choice of vertex-edge assignment on $t_i \in [t_i, t_{i+1}]$ need not be the same as the choice of vertex-edge assignment on $t_i \in [t_{i-1}, t_i]$. This is what we meant by "flexibility" in Definition A.3, the key point is that both choices will give the same RePHINE diagram.

**Proposition A.7.** *Suppose $f_v = g_v = h$ for some vertex coloring function $h$, then*

$$
d_B^{R,0}(\text{RePHINE}(h, f_e)^0, \text{RePHINE}(h, g_e)^0) \leq 2\|f_e - g_e\|_\infty.
$$

*Proof.* Let $h_e^t(x) = (1 - t)f_e(x) + tg_e(x) : X \times X \to \mathbb{R}_{>0}$ be as in the previous proof. Also let $H_e^t : G \to \mathbb{R}_{\geq 0}$ denote the induced function on $G$ in the sense of Definition 2.1. We can divide $[0, 1]$ into finite intervals $[t_0, t_1], [t_1, t_2], ..., [t_n, t_{n+1}]$, where $t_0 = 0, t_{n+1} = 1, t_0 < t_1 < ... < t_{n+1}$, such that for all $t \in [t_i, t_{i+1}]$ and all edges $x, y \in G$, either

$$
H_e^t(x) - H_e^t(y) \leq 0 \text{ or } \geq 0 \quad (\dagger).
$$

For all $s_1, s_2 \in [t_i, t_{i+1}]$, we claim that we can use Definition A.3 to produce the same list of pairs $(v, e)$ (with some flexible adjustments at endpoints if needed).

The death times for the RePHINE diagrams at $s_1$ and $s_2$ may be different. Let us write $d_1^{s_1} \leq ... \leq d_n^{s_1}$ (with multiplicity) to indicate all the finite death times for $s_1$, and similarly we write $d_1^{s_2} \leq ... \leq d_n^{s_2}$ for $s_2$ (with reordering allowed for deaths that occur at the same time). We claim that the corresponding $(v, e)$ produced at $d_1^{s_2}$ and $d_i^{s_2}$ can be chosen to be the same.

1. For each death time that occurs, we are free to choose any of the merging of two components that occurred at that time to be assigned to that death time.

2. At $d_1^{s_1}$, the death occurs between the merging of two vertices $v$ and $w$ by an edge $e$ such that $H_e^{s_1}(e) = d_1^{s_1}$. If $H_e^{s_2}(e) = d_1^{s_2}$, then we can choose the first death to occur with the same edge $e$ between $v$ and $w$.

Otherwise, suppose $H_e^{s_2}(e) > d_1^{s_2}$. There exists an edge $e'$ such that $H_e^{s_2}(e') = d_1^{s_2}$, so $H_e^{s_2}(e) > H_e^{s_2}(e')$. By (†), this means that $H_e^{s_1}(e) \geq H_e^{s_1}(e')$, which implies that $H_e^{s_1}(e') = d_1^{s_1}$. We instead choose the first death in $d_1^{s_1}$ to occur with the edge $e'$ between its adjacent vertices.

In either case, we see that at the first death time, we can choose an assignment such that the RePHINE diagrams at $s_1$ and $s_2$ are merging the same two connected components $a, b$ by the same edge. Now we will show that they will kill off the same vertex. Since $f_v = g_v$, the first comparison will always give the same result. If there is a tie, then we are comparing $f_e$ and $g_e$. Suppose for contradiction, that without loss, $a$ has lower $\gamma$ value at $s_1$ and $b$ has lower $\gamma$ value at $s_2$. This means that there exists an edge $e_a$ adjacent to $a$ such that

$$H_e^{s_1}(e_a) < H_e^{s_1}(e), \text{ for all } e \text{ adjacent to } b.$$

Now, since $b$ has lower $\gamma$ value at $s_2$, this means that there exists an edge $e_b$ adjacent to $b$ such that

$$H_e^{s_2}(e_b) < H_e^{s_2}(e), \text{ for all } e \text{ adjacent to } a.$$

In particular, this means that $H_e^{s_2}(e_b) < H_e^{s_2}(e_a)$ and $H_e^{s_1}(e_b) > H_e^{s_1}(e_a)$, which violates (†). Hence, we will have a consistent vertex to kill off. Finally, if there is a tie, then we flexibly choose the same vertex to kill off. Since we are comparing the same edge, there is a canonical edge associated too.

Thus, we have shown that $d_1^{s_1}$ and $d_2^{s_2}$ can be chosen to give the same vertex edge pair.

3. Inductively, suppose that up to the $i$-th death, both RePHINE diagrams are producing the same pairs and merging the same components.

For the $i + 1$-th death, $d_{i+1}^{s_1}$ occurs between the merging of two connected components $C_1$ and $C_2$ by an edge $e$ such that $H_e^{s_1}(e) = d_{i+1}^{s_1}$. Now if $H_e^{s_2}(e) = d_{i+1}^{s_2}$, then by our inductive hypothesis we can choose both filtration so that they would be merging the same connected components.

Now suppose $H_e^{s_2}(e) \neq d_{i+1}^{s_2}$. By the inductive hypothesis, it cannot be lower, so $H_e^{s_2}(e) > d_{i+1}^{s_2}$. In this case, we look at $d_{i+1}^{s_2}$ itself, which also occurs with an edge $e'$ that merges connected components $C_1'$ and $C_2'$. Hence, we have that

$$H_e^{s_2}(e) > H_e^{s_2}(e') = d_{i+1}^{s_2}.$$

By (†), this means that

$$H_e^{s_1}(e) \geq H_e^{s_2}(e').$$

By our inductive hypothesis, this edge $e'$ cannot have occurred in prior deaths, hence we have that $H_e^{s_2}(e') = d_{i+1}^{s_1}$, and the same arguments as the base case follow through.

In either case, we see that at the $i + 1$-th death time, we can choose an assignment such that the RePHINE diagrams at $s_1$ and $s_2$ are merging the same two connected components $a, b$ by the same edge. Similar to our discussion in the base case, the vertex-edge pair produced would be consistent.

4. After we go through all finite death times, both RePHINE diagrams will have the same list of vertices that are not killed off, which are then matched to real holes.

This proves the claim above. Now, from triangle inequality, we again have that $d_B^{R,0}(\text{RePHINE}(h, f_e)^0, \text{RePHINE}(h, g_e)^0)$ is bounded by the sum

$$\sum_{i=0}^{n} d_B^{R,0}(\text{RePHINE}(h, h_e^{t_i})^0, \text{RePHINE}(h, h_e^{t_{i+1}})^0).$$

For each summand on the right, we assign a bijection with the exact same strategy as the proof of the previous proposition. For the sake of paragraph space, we write $A_i =$

$d_B^{R,0}(\text{RePHINE}(h, h_e^{t_i})^0, \text{RePHINE}(h, h_e^{t_{i+1}})^0)$ and compute that

$$
\begin{aligned}
A_i &= d_B^{R,0}(\text{RePHINE}(h, h_e^{t_i})^0, \text{RePHINE}(h, h_e^{t_{i+1}})^0) \\
&\leq \max_{(v,e)} |d_{t_{i+1}}(v) - d_{t_i}(v)| + |\alpha_{t_{i+1}}(v) - \alpha_{t_i}(v)| + |\gamma_{t_{i+1}}(v) - \gamma_{t_i}(v)| \\
&= \max_{(v,e)} |d_{t_{i+1}}(v) - d_{t_i}(v)| + 0 + |\gamma_{t_{i+1}}(v) - \gamma_{t_i}(v)| \qquad\qquad \text{Since } f_v = g_v \\
&= \max_{(v,e)} |H_e^{t_{i+1}}(e) - H_e^{t_i}(e)| + |\gamma_{t_{i+1}}(v) - \gamma_{t_i}(v)| \\
&\leq (t_{i+1} - t_i)\|f_e - g_e\|_\infty + \max_v |\gamma_{t_{i+1}}(v) - \gamma_{t_i}(v)|.
\end{aligned}
$$

We claim that $|\gamma_{t_{i+1}}(v) - \gamma_{t_i}(v)| \leq \|h_e^{t_{i+1}} - h_e^{t_i}\|_\infty$. Indeed, without loss let us say $\gamma_{t_{i+1}}(v) \geq \gamma_{t_i}(v)$. Let $e_i$ be the edge adjacent to $v$ that has minimum value under $H_e^{t_i}$, then this means that

$$
\begin{aligned}
|\gamma_{t_{i+1}}(v) - \gamma_{t_i}(v)| &= \gamma_{t_{i+1}}(v) - \gamma_{t_i}(v) \\
&= \gamma_{t_{i+1}}(v) - H_e^{t_i}(e_i) \\
&\leq H_e^{t_{i+1}}(e_i) - H_e^{t_i}(e_i) \qquad\qquad \text{Since } \gamma_{t_{i+1}}(v) \text{ is minimum} \\
&\leq \|H_e^{t_{i+1}} - H_e^{t_i}\|_\infty \\
&\leq \|h_e^{t_{i+1}} - h_e^{t_i}\|_\infty \\
&\leq (t_{i+1} - t_i)\|f_e - g_e\|_\infty.
\end{aligned}
$$

Thus, we have that

$$
A_i = d_B^{R,0}(\text{RePHINE}(h, f_e)^0, \text{RePHINE}(h, g_e)^0) \leq 2(t_{i+1} - t_i)\|f_e - g_e\|_\infty.
$$

Hence, we have that

$$
\begin{aligned}
d_B^{R,0}(\text{RePHINE}(h, f_e)^0, \text{RePHINE}(h, g_e)^0) &\leq \sum_{i=0}^{n} A_i \\
&\leq \sum_{i=0}^{n} 2(t_{i+1} - t_i)\|f_e - g_e\|_\infty \\
&= 2\|f_e - g_e\|_\infty.
\end{aligned}
$$

$\square$

Now we finally give a proof of Theorem 4.4.

*Proof of Theorem 4.4.* On the 1-dimensional components of the RePHINE diagram, we have the usual bottleneck distance. Cohen-Steiner et al. [16] gives a standard bound on this term by $d_B(\text{RePHINE}(f)^1, \text{RePHINE}(g)^1) \leq \|f_e - g_e\|_\infty$. The theorem then follows from the triangle inequality, the inequality in the previous sentence, and the previous two propositions. $\square$

Now we show that SpectRe is locally stable under the metric $d_B^{\text{Spec } R}$.

*Proof of Theorem 4.5.* Let us first prove the case with constraints on both $f_v$ and $f_e$. We will again split this into two cases where $f_e = g_e$ and $f_v = g_v$ respectively.

Suppose again that $f_v = g_v = h$, let us try to follow the proof of Proposition A.7 to give an idea on why using this method falls apart. let $h_e^t(x) = (1 - t)f_e(x) + tg_e(x)$ and $H_e^t : G \to \mathbb{R}$ be the induced function of $h_e^t$ on $G$, in the sense of Definition 2.1. We can again divide $[0, 1]$ into finite intervals $[t_0, t_1], [t_1, t_2], ..., [t_n, t_{n+1}]$, where $t_0 = 0, t_{n+1} = 1, t_0 < t_1 < ... < t_{n+1}$, such that for all $t \in [t_i, t_{i+1}]$ and all simplicies $x, y \in G$, either

$$
H_e^t(x) - H_e^t(y) \leq 0 \text{ or } \geq 0 \quad (\dagger).
$$

For all $s_1, s_2 \in [t_0, t_1]$, we can again use Definition A.3 to produce the same list of pairs $(v, e)$ (vertex to edge identification). Previously, by choosing $s_1 = t_0 = 0$ and $s_2 = t_1$, we were able to obtain a reasonable bound on the bottleneck distance for RePHINE in terms of the $L^\infty$ norms of $h_e^{t_0}$ and $h_e^{t_1}$.

We could do this for RePHINE because the $b, d, \alpha, \gamma$ parameters of RePHINE are all not sensitive to the loss of injectivity. However, the $\rho$-parameter in SpectRe is sensitive to the loss of injectivity, as seen in Example 4.6. Moreover, the way we constructed the division of $[0,1]$ indicates that we are forced to cross some time stamps $t$ in $[0,1]$ where $h_e^t$ is no-longer injective.

However, we observe that clearly we could get the desired bound

$$d_B^{\text{Spec } R,0}(\text{SpectRe}(h, f_e)^0, \text{SpectRe}(h, g_e)^0) \leq 2||f_e - g_e||_\infty,$$

provided that the following more restrictive condition holds - $h_e^t$ is injective for all $t \in [0,1]$. The bounds on the $b, d, \alpha, \gamma$ parameters evidently follows from the same proof of Proposition A.5. For the bound of $\rho$, we observe that in the production of the vertex-edge pairs $(v, e)$ in the proof of Proposition A.5, we can choose the order of vertex deaths to be the same for both $(h, f_e) = (h, h_e^0)$ and $(h, g_e) = (h, h_e^1)$. Furthermore, the condition that $h_e^t$ is injective for all $t \in [0,1]$ means that the ordering of colors in $X$ given by $f_e$ and $g_e$ respectively are exactly the same. Furthermore, both orderings are strict as they are injective. Thus, the component that the vertices die in at each time are also the same. What this effectively means is that, $\rho_f(v) = \rho_g(v)$ for all $v \in V$ (after choosing the $(v, e)$ identification). Thus, we would obtain the same bound.

Suppose $f_e = g_e = h$, then we note that an analogous argument (although not required, see the proof of Theorem 4.7 later), would work to show the bound

$$d_B^{\text{Spec } R,0}(\text{SpectRe}(h, f_v)^0, \text{SpectRe}(h, g_v)^0) \leq ||f_v - g_v||_\infty,$$

if we impose the condition that $h_v^t$ is injective for all $t \in [0,1]$ in the context of the proof for Proposition A.5.

We still need to check what happens for $d_B^{\text{Spec } R,1}$, which is no longer the usual bottleneck distance. If $f_e = g_e = h$ and $h_v^t$ is injective for all $t \in [0,1]$, then the 1st dimensional component of $\text{SpectRe}(f_v, h)$ and $\text{SpectRe}(g_v, h)$ would quite literally be identical. If $f_v = g_v = h$, and $h_e^t$ is injective for all $t \in [0,1]$, then a similar argument as in Proposition A.7 would show that

$$d_B^{\text{Spec } R,1}(\text{SpectRe}(h, f_e)^1, \text{SpectRe}(h, g_e)^1) \leq ||f_e - g_e||_\infty.$$

The idea is that the only obstruction to this bound was the presence of the $\rho$-parameter, which we could always choose the presence of cycles to have the same strict order with the same graph components showing up.

Thus, we have proven the following result - let $f = (f_v, f_e)$ and $g = (g_v, g_e)$, suppose $h_v^t(x) = (1-t)f_v(x) + tg_v(x)$ and $h_e^t(x) = (1-t)f_e(x) + tg_e(x)$ are injective for all $t \in [0,1]$, then

$$d_B^{\text{Spec } R}(\text{SpectRe}(f), \text{SpectRe}(g)) \leq 3||f_e - g_e||_\infty + ||f_v - g_v||_\infty.$$

It remains for us to show that the conditions on $h_v^t$ and $h_e^t$ are locally satisfied. However, we note that clearly $\text{Conf}_{n_v}(\mathbb{R})$ and $\text{Conf}_{n_e}(\mathbb{R}_{>0})$ are both locally convex, which is the same as imposing the hypothesis to obtain this bound. Thus, we have proven that $\text{SpectRe}$ is locally stable in $(f_v, f_e)$.

Let us now argue why the local stability condition is not required for $f_v$. Indeed, this is because the parameter $\rho$ in SpectRe is completely independent of the vertex-level filtration. No matter which vertex one decides to kill off, the connected component the vertex dies in is always the same if the edge filtration function does not change (and the same goes for which edge is born). Thus, if $f_e = g_e$, then one can correctly assign the matching of tuples so that the $\rho$-components cancel out identically during a similar linear interpolation proof as in Proposition A.5. This concludes the proof. $\qquad\square$

Now we will give a proof of Theorem 4.7, which gives an upper bound on how unstable SpectRe can be relative to the complexity of the graph.

*Proof of Theorem 4.7.* Let $D$ denote the distance $d_B^{\text{Spec } R,0}(\text{SpectRe}(G, f_v, f_e), \text{SpectRe}(G, g_v, g_e))$. By the explanation in the proof of Theorem 3.4, we know the SpectRe is globally stable in the vertex filtration. Thus, we have that

$$D \leq d_B^{\text{Spec } R,0}(\text{SpectRe}(G, f_v, f_e)^0, \text{SpectRe}(G, g_v, f_e)^0)$$
$$+ d_B^{\text{Spec } R,0}(\text{SpectRe}(G, g_v, f_e)^0, \text{SpectRe}(G, g_v, g_e)^0)$$
$$\leq ||f_v - g_v||_\infty + d_B^{\text{Spec } R,0}(\text{SpectRe}(G, g_v, f_e)^0, \text{SpectRe}(G, g_v, g_e)^0).$$

This means that we have reduced to the case that they have the same vertex-filtration function. Now observe that the SpectRe diagram for both $(G, g_v, f_e)$ and $(G, g_v, f_e)$ are the same outside of the vertex deaths at time $a_i$, $a_{i+1}$ for $f_e$ and time $a_{i+1}$ for $g_e$, by how $g_e$ is constructed. Thus, we can choose a suitable bijection $\pi$ to cancel out the identical pairs outside this critical range. In the critical range, the vertex deaths at time $a_{i+1}$ for $f_e$ are a subset of vertex deaths at time $a_{i+1}$ for $g_e$, such that they have the same $\alpha$ and $\rho$ parameters, so we extend the bijection $\pi$ to match them too. Finally, we match the vertex deaths at time $a_i$ for $f_e$ to the remaining ones left in $a_{i+1}$ for $g_e$. Write $D' = d_B^{\text{Spec } R, 0}(\text{SpectRe}(G, g_v, f_e)^0, \text{SpectRe}(G, g_v, g_e)^0)$, this bijection $\pi$ will give us the bound

$$D' \leq \underbrace{(a_{i+1} - a_i)}_{d\text{-parameter}} + \underbrace{||f_e - g_e||_\infty}_{\gamma\text{-parameter}} + \underbrace{\max_{(H_1, H_2) \in Y} d^{\text{Spec}}(\rho(H_1), \rho(H_2))}_{\rho\text{-parameter}}$$

$$\leq 2(a_{i+1} - a_i) + \max_{(H_1, H_2) \in Y} d^{\text{Spec}}(\rho(H_1), \rho(H_2)).$$

Here, $Y$ is the set indicated in the description of Theorem 4.7, that is - $Y$ is the collection of pairs $(H_1, H_2)$ where $H_1$ is the connected graph that a vertex v died at time $a_i$ for $f_e$ is contained in, $H_2$ is the connected graph the same vertex $v$ died at time $a_{i+1}$ for $g_e$ is in, and $H_1 \subset H_2$. The reason why $Y$ occurs here comes from a direct examination of the graphs $H_1$ and $H_2$ that the $\rho$-parameter considers in the bijection $\pi$.

By the interlacing theorem (Proposition 3.2.1 of Brouwer and Haemers [12]), $d^{\text{Spec}}(\rho(H_1), \rho(H_2))$ can be written as $\text{tr}(\Delta_0(H_2)) - \text{tr}(\Delta_0(H_1))$, where $\text{tr}$ is the matrix trace and $\Delta_0(-)$ is the graph Laplacian. The trace of Laplacian is also the sum of degrees of the graph, which is also twice the number of edges. Thus, we have that

$$D' \leq 2(a_{i+1} - a_i) + 2 \max_{(H_1, H_2) \in Y} |E(H_2)| - |E(H_1)|.$$

This concludes the proof for the 0-dimensional part.

For the 1-dimensional part, we can perform a similar bijection in this case. In this case, the $\alpha$ and $\gamma$ parameters are automatically zero, so the terms they contribute ($||f_v - g_e||_\infty$ for $\alpha$ and $(a_{i+1} - a_i)$ for $\gamma$) also vanish. Following a similar proof as before yields the bound

$$(a_{i+1} - a_i) + 2 \max_{(H_1, H_2) \in Z} |E(H_2)| - |E(H_1)|.$$

$\square$

## B  Expressivity of Spectral Information

In this section, we will give a self-contained proof of Proposition 3.4. We will first introduce the relevant concepts. Suppose $K$ is an $n$-dimensional finite simplicial complex. There is a standard simplicial chain complex of the form

$$\ldots \xrightarrow{\partial_{n+1}} 0 \xrightarrow{} C_n(K) \xrightarrow{\partial_n} \ldots \xrightarrow{} C_1(K) \xrightarrow{\partial_1} C_0(K) \xrightarrow{\partial_0} 0 .$$

Here each $C_i(K)$ has a formal basis being the finite set of $i$-simplicies in $K$, hence there is a way well-defined notion of an adjoint (which is the transpose) $\partial_i^T$ for each $\partial_i$. The $i$-th combinatorial Laplacian of $K$ is defined as

$$\Delta_i(K) = \partial_i^T \circ \partial_i + \partial_{i+1} \circ \partial_{i+1}^T.$$

$\Delta_i$ is a linear operator on $C_i(K)$. Note that when $K$ is a graph and $i = 0$, $\Delta_0(K)$ is exactly the graph Laplacian of $K$. It is a general fact that the dimension of $\ker \Delta_i(K)$ is the same as the $i$-th Betti number of $K$. Hence, the multiplicity of the zero eigenvalues of $\Delta_i(K)$ corresponds to the $i$-th Betti number of $K$. However, the Betti numbers of $K$ (harmonic information) do not give any information on the non-zero eigenvalues of $\Delta_i(K)$ (which we can think of as the non-harmonic information). This is the data that we would like to keep track of.

For a filtration of a simplicial complex $K$ by $\emptyset = K_0 \subseteq K_1 \subseteq \ldots \subseteq K_m = K$, Wang et al. [59] proposed a persistent version of combinatorial Laplacians as follows.

**Definition B.1.** Let $C_q^t = C_q(K_t)$ denote the $q$-th simplicial chain group of $K_t$, $\partial_q^t : C_q(K_t) \to C_{q-1}(K_t)$ be the boundary map on the simplicial subcomplex $K_t$. For $p > 0$, we use $\mathbb{C}_q^{t+p}$ to denote the subset of $C_q^{t+p}$ whose boundary is in $C_{q-1}^t$ (in other words $\mathbb{C}_q^{t+p} := \{\alpha \in C_q^{t+p} \mid \partial_q^{t+p}(\alpha) \in C_{q-1}^t\}$).

We define the operator $\tilde{\partial}_q^{t+p} : \mathbb{C}_q^{t+p} \to C_{q-1}^t$ as the restriction of $\partial_q^{t+p}$ to $\mathbb{C}_q^{t+p}$. From here, we define the $p$-persistent $q$-combinatorial Laplacian $\Delta_q^{t+p}(K) : C_q(K_t) \to C_q(K_t)$ as

$$\Delta_q^{t+p}(K) = \tilde{\partial}_{q+1}^{t+p}(\tilde{\partial}_{q+1}^{t+p})^T + (\partial_q^t)^T \partial_q^t.$$

Note that the multiplicity of the zero eigenvalues in $\Delta_q^{t+p}(K)$ coincides with the $p$-persistent $q$-th Betti number.

Now we will focus on the special case where $K = G$ is a graph. In this case, we only need to look at the $p$-persistent 1-combinatorial Laplacians and the $p$-persistent 0-combinatorial Laplacians. Our goal is to augment the RePHINE diagram, so we intuitively would like to include all the non-zero eigenvalues of the $p$-persistent $q$-combinatorial Laplacians in our augmentation. In this section, we will show that augmentation is no more expressive than simply focusing on the spectral information of the ordinary graph Laplacian.

**Lemma B.2.** *On a graph $G$, the multi-set of non-zero eigenvalues of $\Delta_1(G)$ is the same as the non-zero eigenvalues of $\Delta_0(G)$.*

*Proof.* For ease of notation, we omit the parameter $G$ in the combinatorial Laplacian. Since $G$ has dimension 1, $\partial_0$ and $\partial_2$ are both 0. Hence, the two combinatorial Laplacians may be written as

$$\Delta_0 = \partial_1 \circ \partial_1^T \text{ and } \Delta_1 = \partial_1^T \circ \partial_1.$$

Let $v$ be an eigenvector of $\Delta_0$ corresponding to a non-zero eigenvalue $\lambda$, then

$$\Delta_1(\partial_1^T v) = \partial_1^T(\partial_1 \circ \partial_1^T(v)) = \partial_1^T(\Delta_0(v)) = \partial_1^T(\lambda v) = \lambda(\partial_1^T v).$$

Hence, $\partial_1^T v$ is an eigenvector of $\Delta_1$ with eigenvalue $\lambda$.

We also need to check that if $v, w$ are linearly independent eigenvectors of $\Delta_0$ with the same eigenvalue $\lambda$, then $\partial_1^T v$ and $\partial_1^T w$ are linearly independent. Suppose for contradiction this is not the case, then there exist coefficients $a, b \in \mathbb{R}$ (not all zero) such that

$$0 = a\partial_1^T v + b\partial_1^T w = \partial_1^T(av + bw).$$

This means that $av + bw \in \ker(\partial_1^T) \subset \ker(\Delta_0)$ is a non-zero eigenvector corresponding to the eigenvalue 0. However, we also know that $av + bw$ is a non-zero eigenvector of $\Delta_0$ corresponding to the eigenvalue $\lambda$. Thus, it has to be the case that $av + bw = 0$, so we have a contradiction.

Hence, the non-zero eigenvalues of $\Delta_0$ form a sub-multiset of that of $\Delta_1$. The other direction may also be proven using linear algebra. Alternatively, however, we observe that by the equality of the Euler characteristic,

$$|V(G)| - |E(G)| = \chi(G) = \dim \ker(\Delta_0) - \dim \ker(\Delta_1).$$

Rearranging the terms gives us

$$|V(G)| - \dim \ker(\Delta_0) = |E(G)| - \dim \ker(\Delta_1).$$

This means that $\Delta_1$ and $\Delta_0$ have the same number of non-zero eigenvalues, so their respective multi-sets of non-zero eigenvalues are equal. $\qquad\square$

Let $G$ be a graph and

$$\emptyset = G_0 \subseteq G_1 \subseteq G_2 \subseteq ... \subseteq G_m = G$$

be a sequence of subgraphs of $G$. Recall that $\Delta_q^{t+p}(G)$ denotes the $p$-persistent $q$-combinatorial Laplacian operator. We will first examine what happens when $q = 1$.

**Lemma B.3.** *The 1-combinatorial $p$-persistence Laplacian $\Delta_1^{t+p}(G)$ is equal to $\Delta_1^t(G) = \Delta_1(G_t)$. Moreover, the non-zero eigenvalues of $p$-persistence $\Delta_1^{t+p}(G)$ are the same as the non-zero eigenvalues of $\Delta_0^t(G)$, accounting for multiplicity.*

*Proof.* Recall that $\Delta_q^{t+p}(G)$ is defined as

$$\Delta_q^{t+p}(G) = \eth_{q+1}^{t+p}(\eth_{q+1}^{t+p})^T + (\partial_q^t)^T \partial_q^t.$$

When $q = 1$, we know that $\eth_2^{t+p}(G)$ is the zero matrix since $G$ is a graph, and hence

$$\Delta_1^{t+p}(G) = (\partial_1^t)^T \partial_1^t.$$

This is independent of $p$ and is just $\Delta_1^t(G)$. Finally, from Lemma B.2, we have that $\Delta_1^t(G)$ and $\Delta_0^t(G)$ have the same multi-set of non-zero eigenvalues. $\square$

*Remark* B.4. This is reflective of the definition of the $p$-persistent $k$-th homology group of $G^t$, which is given by

$$H_k^p(G^t) = \ker \partial_k(G^t)/(\operatorname{im} \partial_{k+1}(G^{t+p}) \cap \ker(\partial_k(G^t)))$$

In this case, when $k = 1$, $\partial_{k+1}(G^{t+p}) = \partial_2(G^{t+p})$ is the zero map, so $H_1^p(G^t) = \ker \partial_1(G^t) = H^1(G^t)$. Hence, the $p$-persistent 1st homology groups of $G^t$ stays constant as $p$ varies. This is also reflective of the fact that an inclusion of subgraph $i : G \to G'$ induces an injective homomorphism $i_* : H_1(G) \to H_1(G')$.

The focus of persistent spectral theory should then be on the data given by the graph Laplacians, so it makes sense for us to interpret what exactly $\Delta_0^{t+p}(G)$ is.

**Lemma B.5.** *Suppose* $\mathbb{C}_1^{t+p} = \{\alpha \in C_1^{t+p} \mid \partial_1^{t+p}(\alpha) \in C_0^t\}$ *is equal to the span of all 1-simplicies in $G_{t+p}$ whose vertices are in $G_t$, the $p$-persistent $0$-combinatorial Laplacian operator of $G$*

$$\Delta_0^{t+p}(G) = \eth_1^{t+p}(\eth_1^{t+p})^T$$

*is the graph Laplacian of the subgraph of $G_{t+q}$ with all the vertices in $G_t$.*

*Proof.* The map $\eth_1^{t+p} : \mathbb{C}_1^{t+p} \to C_0^t$ is the restriction map on $\partial^{t+p}$ onto $\mathbb{C}_0^{t+p}$. Let $G'$ denote the subgraph of $G_{t+q}$ generated by vertices in $G_t$. Note that by our assumption $\mathbb{C}_1^{t+p} = C_1(G')$. In this case, there are two vertical isomorphisms, by quite literally the identity map, such that the following diagram commutes,

$$
\begin{array}{ccc}
C_1(G') & \xrightarrow{\partial} & C_0(G') \\
\downarrow & & \downarrow \\
\mathbb{C}_1^{t+p} & \xrightarrow{\eth} & C_0^t
\end{array}
.
$$

Hence, the graph Laplacian of $G'$ is the same as the Laplacian $\Delta_0^{t+p}$. $\square$

**Corollary B.6.** *Lemma B.3 asserts that the non-zero eigenvalues of $\Delta_1^{t+p}(G)$ are the same as the non-zero eigenvalues of $\Delta_0^t(G) = \Delta_0(G_t)$. In the special case where we focus on filtrations of $G$ given by $(F_v, F_e)$ outlined in Section 2, we have that $\Delta_0^{t+p}(G)$ is the same as $\Delta_0(G_{t+p})$ in the edge filtration given by $F_e$.*

*Proof.* We first observe that edge filtrations satisfy the hypothesis of Lemma B.5 since all vertices are spawned at time 0 and hence $C_0^t = C_0^{t+p}$. To prove Corollary B.6, observe Lemma B.5 implies that $\Delta_0^{t+p}(G)$ is the graph Laplacian of the subgraph of $G_{t+p}$ generated by the vertices in $G_t$. However, all vertices are spawned at the start, so they have the same vertex set and the subgraph is just the entire graph $G_{t+p}$. $\square$

Now we will define the alternative descriptor in Proposition 3.4 more formally and prove the theorem.

**Definition B.7.** Given a graph $G = (V, E, c, X)$ and filtration functions $f = (f_v, f_e)$. We define
$$\Phi(f) := \{(b(v), d(v), \alpha(v), \gamma(v), \rho'(v)\}_{v \in V} \sqcup \{(b(e), d(e), \alpha(e), \gamma(v), \rho'(e)\}_{e \in \{\text{ind. cycles}\}}.$$

Here, the first four components are the same as RePHINE (Definition 2.3). The last component $\rho'$ refers to the following data.

Let $d$ be the time the vertex $v$ dies in and $G(v)$ be the connected component of $v$ at time $d$. The edge filtration gives a sub-filtration of $G(v)$ such that at time 0, we start with only the vertices of $G(v)$ and we have all of $G(v)$ at time $d$. This gives a sequence of subgraphs:

$$\text{vertices of } G(v) = G(v)_0 \subsetneq G(v)_1 \subsetneq \ldots \subsetneq G(v)_m = G(v) \quad (\dagger)$$

$\rho'(v)$ returns the eigenvalues of the Laplacians $\Delta_0^{i+p}(G(v))$ and $\Delta_1^{i+p}(G(v))$ for all $i + p = m$ with respect to the filtration $(\dagger)$. Here $i$ refers to the subscript $G(v)_i$ in the filtration. The definition of $\rho'(e)$ for $e$ represents a cycle (in the sense of $H_1(G(v))$ is defined similarly, with the graph being the connected component of $e$ at the time $e$ is born in.

Now we will prove that $\Phi$ is no more expressive than $\mathrm{SpectRe}$ given a technical assumption. The assumption is that we need to remember which vertex is assigned to which tuple and which edge (representing cycle) is assigned to which tuple in our computation of SpectRe. This assumption is always satisfied for a computer algorithm, as computing SpectRe (or RePHINE) in practice already goes through such assignments (see Algorithm 2 of [33]).

*Proof of Proposition 3.4.* Let us look at why the SpectRe alternative $\Phi$ does not give more information than our SpectRe (Def 3.1) for vertices. The argument for the 1st-dimensional components will follow similarly.

By Lemma B.5 and Corollary B.6 above, we have that $\Delta_0^{i+p}(G(v))$ is equal to the graph Laplacian of $G(v)_{i+p} = G(v)_m = G(v)$. On the other hand, Lemma B.3 implies that $\Delta_1^{i+p}(G(v)) = \Delta_1(G(v)_i)$, where $\Delta_1$ denotes the 1st combinatorial Laplacian. The multiplicity of the zero eigenvalues of $\Delta_1^{i+p}(G(v))$ is equal to $\dim H_1(G(v)_i)$, which can be recovered by looking at the number of $(1, d(e))$ with $e \in G(v)$ that has appeared before or at $G(v)_i$.

By Lemma B.3, the non-zero eigenvalues of $\Delta_1^{i+p}(G(v))$ are equal to the non-zero eigenvalues of the graph Laplacian of $G(v)_t$. Although $G(v)_i$ may not be connected, we can mark its connected components as $C_1, \ldots, C_r$ (as graphs). Let $t(i)$ be the time where the filtration $(\dagger)$ got to $G(v)_i$. For each $1 \leq j \leq r$, we let $a_j \leq t(i)$ be the time for which the connected graph $C_j$ is created. The creation of the graph $C_j$ is done either by merging two components (i.e. a vertex in $C_j$ died at time $a_j$) or by adding a cycle (i.e. an $e$ was born at time $a_j$), so there exists a tuple in the SpectRe (Def 3.1) diagram that has the non-zero eigenvalues of the graph Laplacian of $C_j$. Doing this for all $j$ recovers the non-zero eigenvalues of $\Delta_1^{i+p}(G(v))$. $\square$

The upshot is that considering the eigenvalues of persistent Laplacians on edge filtrations reduces to computations that could be found from the graph Laplacian and persistent homology.

## C   RePHINE on Monochromatic Graphs

**Lemma C.1.** *Let G and H be two graphs with the same number of nodes with one single color, then* RePHINE *can differentiate G and H if and only if either $b_0(G) \neq b_0(H)$ or $b_1(G) \neq b_1(H)$.*

Here $b_0, b_1$ refer to the 0th and 1st Betti numbers of G and H, which are the number of connected components and independent cycles.

*Proof.* Since G and H only have one color, $f_v, f_e$ are both constant functions of values $a_V, a_E$, so the $\alpha$ and $\gamma$ parameters of RePHINE for $G$ and $H$ are the same and can be discarded. Here, the filtrations with respect to $f_e$ has two steps - it first spawns all the vertices and then adds in all the edges. Thus, RePHINE can differentiate G and H if and only if looking at its first two parameters alone can differentiate G and H. Since $f_v$ and $f_e$ are constant, a vertex either dies at time $a_V$ or at infinity, and all independent cycles are born at time $a_E$. Thus RePHINE can differentiate G and H if and only if they have different counts of pairs $(0, a_V), (0, \infty)$, and $(1, a_E)$. Since G and H have the same number of vertices, these counts differ if and only if $b_0(G) \neq b_0(H)$ and $b_1(G) \neq b_1(H)$. $\square$

# D   Considerations and Comparisons with Alternative Designs

In this section, we discuss on and how some topological descriptors we described in the main text may behave on alternative filtration and descriptor designs. Specifically, we will be discussing:

1. The behavior of PH for Biparameter Filtrations.
2. The behavior of PH for Vietoris-Rips Filtrations.
3. Spectral information on vertex-based filtrations.

## D.1   Vertex and Edge Filtration vs. Biparameter Filtration

The purpose of this section is to compare the individual filtrations of $f_v$ and $f_e$ against the biparameter filtration inducded by $f_v$ and $f_e$ together.

**Definition D.1.** Let $f, g : G \to \mathbb{R}$ be two filtration functions of a graph $G$. We define a function $f \oplus g : G \to \mathbb{R}^2$ with $f \oplus g(x) = (f(x), g(x))$. A **biparameter filtration** of $G$ is the collection of the subgraphs $G_{s,t} \coloneqq (f \oplus g)^{-1}((-\infty, s] \times (-\infty, t]))$. Let $a_0, ..., a_n$ be the time steps for the filtration of $G$ by $f$, and let $b_0, ..., b_m$ be the time steps for the filtration of $G$ by $g$. We also let $a_{-1} < a_0$ and $b_{-1} < b_0$. The collection $A = \{G_{s,t}\}_{s \in \{a_{-1}, ..., a_n\}, t \in \{b_{-1}, ..., b_m\}}$ and have a poset structure induced by inclusions $G_{s,t} \subset G_{s',t'}$ for $s \le s'$ and $t \le t'$. For our purposes, the **biparameter persistence** of $(G, f, g)$ is the PH of all possible poset pathes in the collection $A$.

The upshot of this section in the appendix is the following result.

**Proposition D.2.** *Let $f_v, f_e : G \to \mathbb{R}$ be vertex-level and edge-level filtration functions (still with respect to a coloring), then the **biparameter persistence** of $(G, f, g)$ is strictly more expressive than the PH of $(G, f)$ added with the PH of $(G, g)$.*

*Proof.* Let $G$ and $H$ be the following graphs:

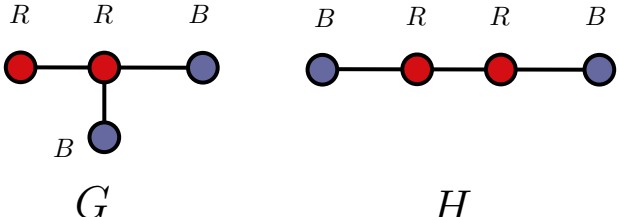

In this case, one can check PH in Definition 2.2 cannot tell them apart for any choices of $f_v$ and $f_e$ (this was also done in [33]). Now let $f_e(R - B) = 1, f_e(R - R) = 2, f_v(B) = 1, f_v(R) = 2$. Consider the sub-filtration of the 2-parameter filtration given by $G_{*,1}$, ie. ($G_{0,1} \to G_{1,1} \to G_{2,1}$.) and $H_{*,1}$ respectively. One can check that this is the vertex-coloring filtration $f_v$ on the subgraphs:

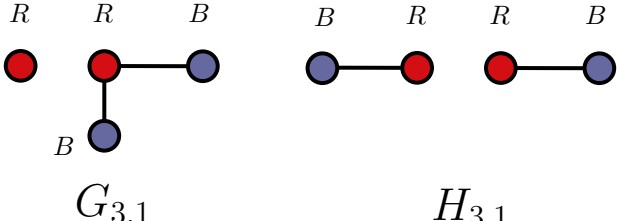

In this case, the left one gives persistence pairs $(1, 2), (2, 2), (2, \infty), (1, \infty)$ (since a red node (R) and a blue node (B) die here at $t = 2$), but the right one gives pairs $(1, \infty), (1, \infty), (2, 2), (2, 2)$ (since the two R's die here at $t = 2$). □

## D.2   Vietoris-Rips Filtrations

In our main paper, we were concerned with conducting filtrations directly on the given graph $G$. The idea of a Vietoris-Rips (VR) filtration is to build a simplicial complex $K$ out of the graph

and examines a filtration on $K$ instead. The hope is that the filtration on $K$ would contain more information. For our purposes, a Vietoris-Rips (VR) filtration is defined as in the discussions right under Theorem 1 of Ballester and Rieck [2]:

**Definition D.3.** Given a graph $G = (V, E)$, we consider the filtration $f_V$ of a simplicial complex $K$, where $K$ is the set of all non-empty subsets $S$ of $V$ such that the diameter of $S$ is not infinity (ie. they are all reachable from one another), and the filtration is $f_V(S) = max_{u,v}d_G(u,v)$ (where $d_G$ is the shortest-path distance). We write $\mathrm{VRPH}(G)$ to be the persistence diagrams associated to this filtration $(K, f_V)$.

We note here that VRPH is, in general, incomparable to PH (Definition 2.2) in terms of expressive power. Indeed, let $G$ be a path-graph on 3 vertices and $H$ be a cycle graph on 3 vertices, then VRPH cannot differ them (see the discussions right before Section 5 of [2]), but PH clearly can, due to the presence of a cycle in H that it can detect. On the other hand, the PH used in Definition 2.2 is color-based and processes graphs with the structure of colors on them. On monochromatic graphs, PH is no more expressive than the number of connected components and independent cycles, whereas VRPH can tell more examples apart as it does not need to respect the colors. More comparisons between VRPH and other persistence methods are present in Table 3 of Ballester and Rieck [2].

Thus, rather than viewing VR filtrations as a strictly richer simplicial filtration, the graph filtration we considered and the VR filtrations are really complementary to each other. There is no reason to expect the two to be comparable if we also add in spectral information either. It would be interesting, though, to look at how combinatorial Laplacians and persistence Laplacians method behave on VR filtrations (or simplicial filtrations in general).

We do note that, if a simplicial filtration of $K$ adds in the entire graph first, and then adds in the higher dimensional simplices at a later time, then it does subsume the context of a graph-based filtration. A full simplicial filtration may in general be quite difficult to enumerate when the scale of the base graph gets larger. For example: on a connected graph $G$ with $n$ vertices, the full VR filtration on $G$ will go through $2^n - 1$ many simplices. This is exponential in scale whereas SpectRe can be done in polynomial time with respect to $G$.

### D.3 Adding Vertex-level Spectral Information to SpectRe

In the definition of SpectRe (Definition 3.1), we added an extra $\gamma$-parameter with respect to the edge-level filtration $f_e$. One can observer, as in the proof of Theorem 4.5, that changing the function $f_v$ has no effect on the $\rho$-parameter of the respective vertices. One can consider what would happen if we want to add spectral information with respect to $f_v$ as well.

From here we make an interesting observation on whether Proposition 3.4 extends to the case of $f_v$. For edge-based $f_e$, we observe that the proof of Proposition 3.4 follows from Corollary B.6 in Appendix B - that the non-zero eigenvalues of persistent Laplacians in edge-based filtrations on graphs can be recovered by the eigenvalues of the graph Laplacians. We note this is however not true for vertex-based filtrations.

Indeed, consider a two step filtration $K \subset L$ where $L$ is the path-graph on 4-vertices labeled 1-2-3-4 and $K$ is the discrete subgraph $\{1, 4\}$. This filtration is the vertex-based filtration of a function $f_v$ given by $f_v(1) = f_v(4) = 1$ and $f_v(2) = f_v(3) = 2$. If we only look at the graph Laplacian spectra of the filtration, we would get that $K$ has eigenvalues $0, 0$ and $L$ has eigenvalues $0, 2 - \sqrt{2}, 2, 2 + \sqrt{2}$.

The persistent 0-dim Laplacian of the pair $(K, L)$ is the matrix $\begin{pmatrix} 1/3 & -1/3 \\ -1/3 & 1/3 \end{pmatrix}$ with eigenvalues $0, 2/3$. This can be verified using the Matlab code in Mémoli et al. [44] with inputs

```
B1 = [0 0];
B2 = [-1 0 0; 1 -1 0; 0 1 -1; 0 0 1];
Gind = [1 4];
```

The extra $2/3$ cannot be recovered from the graph Laplacian spectra of $K$ and $L$ alone. Indeed, for a different filtration of $L$ with $K' = \{1, 3\}$, we would get the matrix $\begin{pmatrix} 1/2 & -1/2 \\ -1/2 & 1/2 \end{pmatrix}$ with eigenvalues $0, 1$.

# E    Datasets and implementation details

Table 4: Statistics of datasets for graph classification, for TUDatasets we obtain a random 80%/10%/10% (train/val/test) split. For ZINC and OGB-MOLHIV, we use public splits.

| Dataset | #graphs | #classes | Avg #nodes | Avg #edges | Train% | Val% | Test% |
|---|---|---|---|---|---|---|---|
| MUTAG | 188 | 2 | 17.93 | 19.79 | 80 | 10 | 10 |
| PTC-MM | 336 | 2 | 13.97 | 14.32 | 80 | 10 | 10 |
| PTC-MR | 344 | 2 | 14.29 | 14.69 | 80 | 10 | 10 |
| PTC-FR | 351 | 2 | 14.56 | 15.00 | 80 | 10 | 10 |
| NCI1 | 4110 | 2 | 29.87 | 32.30 | 80 | 10 | 10 |
| NCI109 | 4127 | 2 | 29.68 | 32.13 | 80 | 10 | 10 |
| IMDB-B | 1000 | 2 | 19.77 | 96.53 | 80 | 10 | 10 |
| MOLHIV | 41127 | 2 | 25.5 | 27.5 | Public Split | | |
| ZINC | 12000 | - | 23.16 | 49.83 | Public Split | | |

**Datasets.**    Table 4 reports summary statistics of the real-world datasets used in the paper. MUTAG contains 188 aromatic and heteroaromatic nitro compounds tested for mutagenicity with avg. number of nodes and edges equal to 17.93 and 19.79, respectively. The PTC dataset contains compounds labeled according to carcinogenicity on rodents divided into male mice (MM), male rats (MR), female mice (FM) and female rats (FR). For instance, the PTC-MM dataset comprises 336 graphs with 13.97 nodes (average) and 14.32 edges (average). Except for ZINC and MOLHIV, all datasets are part of the TUDataset repository, a vast collection of datasets commonly used for evaluating graph kernel methods and GNNs. The datasets are available at `https://chrsmrrs.github.io/datasets/docs/datasets/`. In addition, MOLHIV is the largest dataset (over 41K graphs) and is part of the Open Graph Benchmark[1]. We also consider a regression task using the ZINC dataset — a subset of the popular ZINC-250K chemical compounds [34], which is particularly suitable for molecular property prediction [21].

Our first set of synthetic datasets comprises minimal Cayley graphs — a special class of Cayley graphs only partially understood. For instance, it is unkonwn whether their chromatic number is bounded by a global constant. These datasets have been used to assess the expressivity of graph models [2] and can be found at `https://houseofgraphs.org/meta-directory/minimal-cayley` . BREC is a benchmark for GNN expressiveness comparison. It includes 800 non-isomorphic graphs arranged in a pairwise manner to construct 400 pairs in four categories (Basic, Regular, Extension, CFI). Basic graphs consist of 60 pairs of 1-WL-indistinguishable graphs. Regular graphs consist of 140 pairs of regular graphs split into simple regular graphs, strongly regular graphs, 4-vertex condition graphs and distance regular graphs. For further details on the remaining graph structures, we refer to Wang and Zhang [61].

**Experimental setup.**    We implement all models using the PyTorch Geometric Library [23]. Our implementation is an extension of the official code repository of [33]. For all experiments, we use a cluster with Nvidia V100 GPUs.

For the experiments on real data, we employ MLPs to obtain vertex and edge filtrations followed by sigmoid activation functions, following [33]. We use two different DeepSets to process the 0-dim and 1-dim diagrams.

Regarding model selection, we apply grid-search considering a combination of $\{1, 2\}$ GNN layers and $\{1, 4\}$ filtration functions. We set the number of hidden units in the `DeepSet` and GNN layers to 32, and of the filtration functions to 16 — i.e., the vertex/edge filtration functions consist of a 2-layer MLP with 16 hidden units. The GNN node embeddings are combined using a global mean pooling layer. We employ the Adam optimizer [35] with a maximum of 500 epochs, learning rate of $10^{-4}$, and batch size equal to 64.

We use a random 80%/10%/10% (train/val/test) split for all datasets. All models are initialized with a learning rate of $10^{-3}$ that is halved if the validation loss does not improve over 10 epochs. We apply early stopping with patience equal to 30.

---

[1] `https://ogb.stanford.edu`

**Implementation note.** For the sake of full transparency, while preparing the final version of the manuscript, we discovered that an earlier version of our implementation contained a minor error in the computation of the augmented Forman–Ricci curvature. This issue affected a few results reported in Table 2 and Table 5, and accounts for small numerical discrepancies with the previous version of the manuscript. After correcting the implementation, all results were recomputed. Importantly, the correction does not alter the qualitative findings of the paper: in particular, the conclusion that SpectRe is the best-performing model across our experiments remains unchanged. Further details are available in the official code repository.

# F  Additional experiments

For completeness, we consider two sets of additional experiments. First, we run an ablation study to measure the impact of using partial information on SpectRe's performance on BREC datasets. The second group of experiments aims to assess the performance of SpectRe when combined with other graph neural networks. To do so, we consider the graph transformer model in [48] as backbone GNN.

Table 5 shows results regarding SpectRe using partial spectrum information (one third of the total eigenvalues). As we can see, SpectRe with partial spectrum can distinguish the same graphs as the full spectral approach on Basic, Regular, and Extension. However, if we remove the spectral information, the expressivity drops significantly — SpectRe without spectrum reduces to RePHINE. We note that using partial spectrum is one approach to speed up SpectRe.

Table 5: **Additional ablation study**: SpectRe with partial spectral information (1/3 of the total number of eigenvalues) and Laplacian Spectrum (LS) on the BREC datasets. The results show that using only a small subset of eigenvalues allows distinguishing *almost* as many graphs as the original (full spectrum) approach. Note that RePHINE corresponds to SpectRe with no spectral information.

| Dataset | $PH^0$ | $PH^1$ | RePHINE | LS | SpectRe | SpectRe (partial spectrum) |
|---|---|---|---|---|---|---|
| Basic (60) | 0.03 | 0.98 | 0.98 | **1.00** | **1.00** | **1.00** |
| Regular (50) | 0.00 | 0.94 | 0.94 | **1.00** | **1.00** | **1.00** |
| Extension (100) | 0.07 | 0.55 | 0.55 | **1.00** | **1.00** | **1.00** |
| CFI (100) | 0.03 | 0.03 | 0.03 | **0.04** | **0.04** | 0.03 |
| Distance (20) | 0.00 | 0.00 | 0.00 | **0.05** | **0.05** | 0.00 |

Table 5 also reports results using LS (Laplacian Spectrum) on BREC datasets. The results show that LS perform on par with SpectRe on all datasets. This is somehow expected, given the identical performance of $PH^1$ and RePHINE under our simple choice of filtrations. It is worth noting that LS is a simplified version of SpectRe, and is also a contribution of this paper — as far as we know, no prior work has exploited persistent spectral information in graph learning.

Table 6 shows the results of integrating SpectRe into GPS. Leveraging topological descriptors boosts the performance of the graph Transformer in 3 out of 4 datasets. Again the gains achieved by SpectRe are higher than those obtained with RePHINE. In these experiments, we applied the fast variant of SpectRe: specifically, we used the power method to approximate the largest eigenvalue when $n > 9$, and computed the full spectrum otherwise. Additionally, we employed a scheduling strategy, computing the eigendecomposition at only one-third of the filtration steps.

Table 6: **Graph Transformer (GPS, [48]) and SpectRe**. Here, we consider the combination of topological descriptors with a SOTA graph model. As we can observe, SpectRe boosts the performance of the GPS model and beats RePHINE. For ZINC, we only considered a single filtration.

| Method | NCI1 | NCI109 | IMDB-BINARY | ZINC |
|---|---|---|---|---|
| GPS | $81.51 \pm 1.72$ | $77.00 \pm 0.68$ | **76.00** $\pm 2.83$ | $0.38 \pm 0.01$ |
| GPS+RePHINE | $82.36 \pm 0.86$ | $77.97 \pm 2.74$ | $71.50 \pm 2.12$ | **0.34** $\pm 0.04$ |
| GPS+FastSpectRe | **83.33** $\pm 2.23$ | **79.66** $\pm 0.34$ | $75.50 \pm 0.71$ | **0.34** $\pm 0.02$ |

Finally, we also provide experimental results regarding our stability bounds. In particular, we look at the first 4 graphs from the BREC dataset (basic.npy). For each graph, we define the base filtrations $(f_v, f_e)$ with $f_v$ being the degree of the vertex and $f_e$ being the average of the degree of the two vertices. Table 7 shows the bottleneck distance and the inequality bound $3||f_e - g_e|| + ||f_v - g_v||$ for different choices of $(g_v, g_e)$.

Table 7: **Empirical validation of the stability bounds**.

| $g_v$ | $g_e$ | **Graph ID** | **RePHINE Dist** | **RePHINE Bound** | **SpectRe Dist** | **SpectRe Bound** |
|-------|-------|--------------|------------------|-------------------|------------------|-------------------|
| $\sin(10f_v)$ | $5f_e$ | G1 | 65.25 | 78.30 | 66.23 | 78.30 |
| | | G2 | 55.25 | 66.30 | 56.30 | 66.30 |
| | | G3 | 68.25 | 86.99 | 69.25 | 86.99 |
| | | G4 | 62.26 | 72.30 | 62.26 | 72.30 |
| $0.1f_v + 0.1$ | $\exp(-f_e)$ | G1 | 21.18 | 24.19 | 71.95 | 24.19 |
| | | G2 | 17.77 | 20.28 | 65.98 | 20.28 |
| | | G3 | 22.59 | 26.60 | 72.37 | 26.60 |
| | | G4 | 18.78 | 21.79 | 60.60 | 21.79 |

The results in Table 7 correspond to the expected behaviors. We first see that no matter what the $(g_v, g_e)$ is, the RePHINE distance is always bounded by the RePHINE bound, which corresponds to RePHINE being *globally stable*. For SpectRe, setting $g_v = \sin(10f_v), g_e = 5f_e$ does not break stability. This is because of two reasons (1) changes in the vertex-filtration function do not affect stability, and (2) the change $f_e \to 5g_e$ does not cross the region of non-injectivity. We also remark that Reason (2) shows there is a great flexibility to perturb $f_e$ without breaking stability. When $g_v = 0.1f_v + 0.1, g_e = \exp(-f_e)$, we see the stability of SpectRe is broken. This is expected behavior because $\exp(-f_e)$ is an order reversing function, and the path from $f_e$ to $\exp(-f_e)$ would necessarily cross some region of non-injectivity.

