# OpenReview forum: "Graph Persistence goes Spectral"
_NeurIPS.cc/2025/Conference — NeurIPS 2025 poster_

### Official Review · Reviewer_MSD2 · 2025-06-28

**Clarity:** 3
**Significance:** 3
**Originality:** 3
**Rating:** 5
**Confidence:** 4

**Summary:**

In this paper, authors propose SpectRe, a novel topological descriptor for graphs that incorporates spectral information into persistent homology (PH) diagrams. SpectRe offers greater expressiveness compared to existing graph descriptors. The authors also present concepts of global and local stability to evaluate existing methods, demonstrating that SpectRe exhibits local stability. Experimental results  showcase the effectiveness of SpectRe and highlight its potential to enhance the performance of graph-based models in various learning tasks.

**Questions:**

1. In the paper, authors mentioned "Homology captures harmonic information (in the sense that the kernel of the graph Laplacian corresponds to the 0-th homology), but there is some non-harmonic information we also want to account for..." I am not so sure if harmonic information is necessary? Could you please explain a bit?
2. The experiments provided are insufficient to fully support the authors' claims. For instance, it is unclear whether the authors considered comparisons with local persistent homology methods.
3. Could you please clarify the steps involved in the procedure described in Section 5 (Integration with GNNs)?

**Ethical Concerns:**

["NO or VERY MINOR ethics concerns only"]

**Final Justification:**

The authors make it clear that the main motivation and contribution. Moreover, it shows it claim, particularly the theorecial part, how to contribute to the community.

**Limitations:**

Yes

**Quality:**

3

**Strengths And Weaknesses:**

Strengths:
1. The authors demonstrate that the expressiveness provided by the non-zero eigenvalues of these spectral descriptors can be equivalently achieved by incorporating the non-zero eigenvalues of the graph Laplacians at all time steps.
2. The authors conduct further theoretical analysis of SpectRe.
3. The authors perform experiments, and the results highlight SpectRe's enhanced expressiveness while demonstrating its potential to improve the performance of graph neural networks on related tasks.

Weaknesses:
1. Overall, this is a strong paper. However, the introduction could be slightly improved by being more concise and clearly articulating the motivation and contributions, particularly for researchers without a background in topological data analysis (TDA).
2. The work heavily relies on results from previous studies, which limits the novelty of its theoretical contributions.
3. The experiments provided are insufficient to fully support the authors' claims. For instance, it is unclear whether the authors considered comparisons with local persistent homology methods.

---

> ### Author Rebuttal · Authors · 2025-07-31
>
> Thank you for your suggestions. We reply to your comments below.
>
> ```
> Introduction: motivation and contributions
> ```
> Thank you for your suggestion. We will edit the introduction accordingly in the paper. To clarify our motivation and contribution: persistent homology (PH) based methods have been used in GNNs to boost the expressivity beyond the WL hierarchy. The spectra of (combinatorial) Laplacians can be seen as a generalization of homology, as homology corresponds to their kernels (see our response to the **harmonic information** question below for details), and thus we are motivated to include Laplacians as a next step. We chose RePHINE as our base to build SpectRe on, but, in principle, our augmentation would apply to any PH-based methods. Pleease see our response right below for a detailed account of our theoretical contributions.
>
> ```
> The work heavily relies on results from previous studies, which limits the novelty of its theoretical contributions.
> ```
> Thank you for giving us the opportunity to clarify our work. Although we chose RePHINE as the base descriptor to incorporate spectral information, we could have created a similar construction with any persistent based descriptors in the given filtration. We chose RePHINE as our base because it is a simple topological descriptor that is provably more expressive than traditional PH. Otherwise, the Laplacian spectrum (LS) (defined after Def 3.1) could have been a more generic option, but its expressivity is bounded by SpectRe.
>
> For Section 3, **Theorem 3.4** shows that augmenting RePHINE with the non-zero eigenvalues of graph Laplacians is just as expressive as augmenting with the entire family of persistent Laplacian spectra outlined in [1]. Using essentially the proof of Theorem 3.4 in the appendix, the same proof holds for augmenting any descriptor that is at least as expressive as PH. In other words, for any descriptor X such that $X \succeq PH$, augmenting X with the non-zero eigenvalues of graph Laplacians has the same expressive power as adding the entire family of persistent Laplacian spectra.
>
> For Section 4, we first want to emphasize that the (global) stability of RePHINE (**Theorem 4.4**) was an open question in the RePHINE paper that we resolved here, which also led to our proof of a stability result for SpectRe (**Theorem 4.5**). There were three difficulties in proving stability properties for RePHINE and SpectRe that prevented us from directly using previous studies (ie. [2]). For RePHINE alone, the original definition of Bottleneck distance did not generalize to RePHINE diagrams as it did not take into account the $\alpha$ and $\gamma$ parameters. To address this, we needed to come up with a suitable version of Bottleneck distance for RePHINE (and SpectRe). The second is that the original proof for Bottleneck stability is not applicable to our setting because it does not account for the $\alpha$ and $\gamma$ parameters. In the appendix, to prove **Theorem 4.4**, we had to come up with the concept of a `vertex-edge pairing' in **Definition A.3**. Finally, for SpectRe, we even lose the global stability of PH and RePHINE as seen in Example 4.6. In order to quantify a weaker notion of stability, we used configuration spaces (ie. Conf_n(R)) to describe a local version of stability to correctly describe the stability of SpectRe. These are theoretical challenges we needed to overcome that suggest the stability section is not a direct application of previous studies. We also want to note that the ``vertex-edge pairing" in Def A.3 and the proofs can give approaches to proving the stability of persistent descriptors whose tuple size is greater than 2.
>
> ```
> The experiments provided are insufficient to fully support the authors' claims.
> ```
> Thanks for the opportunity to elaborate on our experiments. We conducted three sets of experiments: (1) Expressivity results on multiple datasets (Table 2), (2) experiments on graph classification (Tables 3 and 6), and (3) ablation studies (Table 5). We believe these experiments demonstrate the effectiveness of SpectRe (and its fast variant) and are aligned with the claims in the paper.
>
> We have additionally run experiments to showcase the global and local stability of RePHINE and SpectRe. We look at 4 graphs from BREC (basic.npy). For each one, we define ($f_v$, $f_e$) with $f_v$ being the degree and $f_e$ being the average of the degree of the vertices. The following table shows the bottleneck distance and our bound for different choices of $(g_v, g_e)$.
>
> | $g_v$              | $g_e$              | **Graph ID** | **RePHINE Dist** | **RePHINE Bound** | **SpectRe Dist** | **SpectRe Bound** |
> |--------------------|--------------------|--------------|------------------|-------------------|------------------|-------------------|
> | $\sin(10 f_v)$     | $5 f_e$            | G1           | 65.25            | 78.30             | 66.23            | 78.30             |
> |                    |                    | G2           | 55.25            | 66.30             | 56.30            | 66.30             |
> |                    |                    | G3           | 68.25            | 86.99             | 69.25            | 86.99             |
> |                    |                    | G4           | 62.26            | 72.30             | 62.26            | 72.30             |
> | $0.1f_v+0.1$       | $\exp(-f_e)$       | G1           | 21.18            | 24.19             | 71.95            | 24.19             |
> |                    |                    | G2           | 17.77            | 20.28             | 65.98            | 20.28             |
> |                    |                    | G3           | 22.59            | 26.60             | 72.37            | 26.60             |
> |                    |                    | G4           | 18.78            | 21.79             | 60.60            | 21.79             |
>
> We first see that no matter what the $(g_v, g_e)$ is, the RePHINE distance is bounded by the RePHINE bound, which corresponds to RePHINE being **globally stable**.
>
> For SpectRe, when $g_v = \sin(10 f_v), g_e = 5 f_e$, we observe that we did not break stability. This is because of two reasons (1) changes in the vertex-filtration function do not affect stability, and (2) the change $f_e->5g_e$ did not cross the region of non-injectivity. When $g_v = 0.1f_v+0.1, g_e=\exp(-f_e)$, we see the stability of SpectRe is broken. This is expected behavior because $\exp(-f_e)$ is an order reversing function, and the path from $f_e$ to $\exp(-f_e)$ would necessarily cross some region of non-injectivity.
>
> ```
> Comparison with local persistent homology methods.
> ```
> Thank you for raising this question. Given a fixed filtration method, the local PH method in [3] takes in a graph G with no labels on its vertices and produces a map plh_p: V(G) -> R^d where for each vertex v, we takes an appropriate annular local subgraph around v and calculate the PH of the filtration method applied to the subgraph, and then `vectorize' the PH diagram. In the paper, the authors considered the Vietoris-Rips filtration, which produces a simplicial complex with a filtration from the subgraph, so its PH has higher dimensions at the expense of needing to do more calculations. On the other hand, the PH in our work only concerns 0th and 1st dim information. We do note, however, that our method would subsume local PH if we replace the filtration method by a vertex-based or edge-based filtration on the annular local subgraph itself. This is because we can always choose a convenient coloring function such that it repeats the filtration for the annular local subgraph first before loading in the other parts of the graph.
>
> ```
> I am not so sure if harmonic information is necessary?
> ```
> Thank you for raising this question. To clarify, the graph Laplacian and more generally combinatorial Laplacians are usually thought of as the discrete analog of Laplacians in the calculus case. In calculus, solutions to $\Delta u = 0$ are called **harmonic functions**, and hence the kernel of the combinatorial Laplacians are sometimes referred to as **harmonic information**. When we extend this to persistent Laplacians, the kernel of the persistent Laplacians (ie. the harmonic information) capture precisely the persistent homology (see the introduction and Section 3 of [1]). This is a generalization of the fact that the dimension of the graph Laplacian's kernel is the same as the number of connected components (ie. the 0th homology). Therefore, harmonic information is quite necessary for our studies because they are exactly persistent homologies. In the design of SpectRe, we only needed to include the non-zero eigenvalues precisely because persistent homology already accounted for the zero ones.
>
> ```
> On the integration with GNNs
> ```
> To integrate SpectRe into GNNs, we first need to obtain a vector representation of the persistence diagrams. This is done in two stages:
>
> 1) We use a DeepSet to encode the graph spectrum associated with each tuple in the diagram.
> 2) A second DeepSet is then applied to aggregate the encoded tuples into a single vector, yielding a fixed-size representation of the diagram.
>
> At each GNN layer, we compute a new SpectRe diagram and derive its corresponding vector representation. These representations are then integrated into the model—for example, by concatenating them with the graph-level representation—just before the final classification head. We will clarify this process in the revised manuscript.
>
> ---
> We hope your concerns have been satisfactorily addressed, and if so, would appreciate if you could revisit your score to reflect the same. We are also committed to engaging further if you have any additional concerns.
>
> [1] Persistent Topological Laplacians – a Survey. X. Wei and G-W. Wei
>
> [2] Stability of Persistence Diagrams. D. Cohen-Steiner, H. Edelsbrunner, J. Harer
>
> [3] Persistent Local Homology in Graph Learning. M. Wang, Y. Hu, Z. Huang, D. Wang, J. Xu

---

> > ### Comment · Reviewer_MSD2 · 2025-08-03
> >
> > Thanks for the authors responds. It solves my concerns. I think it is a good paper and I will revise my rate.

---

> > > ### Author Response · Authors · 2025-08-03
> > >
> > > Thank you for your positive response! We are glad to hear that your concerns have been addressed, and we will for sure incorporate your helpful feedbacks in the final version.

---

### Official Review · Reviewer_77B6 · 2025-07-03

**Clarity:** 3
**Significance:** 2
**Originality:** 3
**Rating:** 4
**Confidence:** 4

**Summary:**

This paper introduces a new topological descriptor for graphs, SpectRe, which enhances RePHINE diagrams by incorporating the nonzero eigenvalues of the graph Laplacian at each filtration step. The authors show that SpectRe is strictly more expressive than both RePHINE and the Laplacian spectrum alone, and establish that it is invariant under graph isomorphisms. They also define custom bottleneck-like metrics under which RePHINE is globally stable and SpectRe is locally stable (under an injectivity assumption). Experimental results on graph isomorphism tasks (Cayley graphs, BREC) and real-world GNN benchmarks (MUTAG, ZINC, etc.) show that SpectRe outperforms prior persistence-based descriptors in both discriminative and predictive settings.

**Questions:**

1. How sensitive is SpectRe to noise or perturbations in the filtration functions, particularly when injectivity is violated?

2. Can the stability results be extended to include non-injective filtrations in some approximate sense?

3. Would using approximate spectral methods (e.g., Nyström) change expressivity, or just reduce fidelity?

**Ethical Concerns:**

["NO or VERY MINOR ethics concerns only"]

**Final Justification:**

This paper introduces a well-motivated and clearly beneficial modification of persistence-based graph descriptors. The improvements over RePHINE and the Laplacian spectrum are theoretically well-justified. However, downstream improvements in Table 3 are modest and in many cases not clearly statistically significant. Additionally, after the rebuttal discussion, I note that the additional experiments in the appendix (e.g., Graph Transformers, LS on BREC) help clarify the relative contributions of each component and this goes part of the way toward addressing my request for ablations. The expanded discussion on stability and the injectivity assumption in the rebuttal was useful and I appreciate the formal attempt to quantify the impact of injectivity violations; including this in the main text would strengthen the paper. All things considered, I now lean towards a weak accept rating for the paper and have updated my review accordingly.

**Limitations:**

The authors note the computational complexity and the lack of global stability for SpectRe. I’d add that the injectivity assumption may be hard to guarantee in real applications, and there’s some ambiguity about how robust SpectRe is to near-ties or small perturbations in filtration values. The method also assumes access to full eigenvalue spectra (at least at small graph sizes), which may limit applicability at larger scale.

**Paper Formatting Concerns:**

Formatting is mostly fine. Some figures (especially Figure 2) are hard to read due to small fonts. The notation in Section 4 gets heavy and could benefit from a summary table or inline simplification.

**Quality:**

3

**Strengths And Weaknesses:**

## Strengths

1. The idea is conceptually clean and fills a real gap in the PH literature. Augmenting persistence diagrams with spectral information has been discussed before, but this is the first implementation that (i) actually improves expressivity over RePHINE and Laplacian individually and (ii) is formalized with a proper stability framework.

2. The theoretical results are solid. The expressivity separation is clearly demonstrated (Figure 3), and the stability results (global for RePHINE, local for SpectRe) are carefully stated and proven with new metrics defined for each case. The counterexample in Example 4.6 is also good for setting expectations.

## Weaknesses

1. The method is significantly more expensive than standard PH descriptors, especially on large or dense graphs. The authors mention scheduling and power methods, but they don’t provide any concrete timing comparisons or resource benchmarks.

2. I do not find the downstream empirical performance on real data to be particularly convincing. For most of the datasets in Table 3, the separation between SpectRe and RePHINE does not seem to be statistically significant. The mean improves slightly, but the standard deviation is rather high in most cases.

3. The stability of SpectRe only holds locally under injectivity, which is a strong assumption. Many real-world filters (e.g., curvature-based or learned ones) may not satisfy this. It would be good to have some robustness analysis or at least an empirical sanity check here.

4. The lack of ablation is noticeable. It’s not clear whether the gains come primarily from the spectral component or whether there is some synergy with RePHINE. A comparison with a "spectrum-only" variant would help clarify this.

## Verdict

This paper introduces a well-motivated and clearly beneficial modification of persistence-based graph descriptors. The improvements over RePHINE and the Laplacian spectrum are theoretically well-justified. That being said, Table 3 does not leave me with a particularly convincing impression of the real-world empirical utility of the method and I do think the lack of ablations and runtime analysis weakens the empirical section. Additionally, the local-only stability is a limitation worth highlighting. All things considered, I am inclined to give the paper a weak reject rating in its current state.

---

> ### Author Rebuttal · Authors · 2025-07-31
>
> Thank you for your comments. We reply to your comments/questions below.
>
> ```
> The method is significantly more expensive than standard PH descriptors, ...
> ```
> Thank you for your comment. The average times per epoch (measured over 10 epochs) on CPU for FastSpectre (with scheduling) on the largest datasets are as follows: NCI1 / NCI109 (11s), IMDB-BINARY (10s), ZINC (17s), and MOLHIV (130s). This is substantially faster than the full method, which requires NCI1 / NCI109 (26s), IMDB-BINARY (24s), ZINC (40s), and MOLHIV (211s). As discussed in the paper, the faster variants of SpectRe can be applied to larger datasets while still offering greater expressivity compared to other topological graph descriptors.
>
> ```
> I do not find the downstream empirical performance on real data to be particularly convincing. For most of the datasets in Table 3, the separation between SpectRe and RePHINE does not seem to be statistically significant...
> ```
> We note that the gains over RePHINE are consistent across datasets and GNNs (see Table 6 for experiments using Graph Transformers). Overall, these results indicate that when combining topological descriptors with GNNs on attributed datasets (with informative node features), the latter’s inductive biases play a significant role. Similar findings have also been observed, for instance, in Topological GNNs (ICLR, 2022).
>
> Most important, our expressivity results are significant and clearly demonstrates benefits of our proposal over existing topological descriptors, which is the primary motivation for our proposal.
>
> ```
> The stability of SpectRe only holds locally under injectivity, which is a strong assumption. Many real-world filters (e.g., curvature-based or learned ones) may not satisfy this. ... have some robustness analysis or ... empirical sanity
> ```
> Thank you for raising this! From the issue pointed out on injectivity, we came up with an approximate bound on how worse the error could get when injectivity is violated and some extensions of the stability results (see our response to Q1 & Q2 below).
>
> Here, we note that our stability result holds on filtration functions that are ``injective on the level of colors". In other words, the filtration function produces the same value on v and w if and only if v and w have the same color. The curvature-based scheme in [2] assigns filtrations to graphs without labels (ie. colors), which is somewhat different. In our context, if we are given a graph without labels, we typically would assign it some form of colors first in order to be able to apply the theorem. The curvature-based method in [2] does not really adapt to give a filtration for an arbitrary graph with some colors.
>
> We also note that the injectivity assumption is reasonable in many cases. For example, if fv: X -> R is not injective. Then the filtration on the graph is the same as the filtration induced by an injective function fv': X' -> R where X' is X/~ where we say two colors a, b are equivalent if fv(a) = fv(b).
>
> We also did an empirical verification of the stability results. Please see our response to **Reviewer jmrk** for more details
>
> ```
> The lack of ablation is noticeable. ... A comparison with a "spectrum-only" variant would help clarify this.
> ```
> We report in Table 5 of the Appendix additional experiments considering SpectRe using partial spectrum information (one third of the total eigenvalues). As we can see, SpectRe with partial spectrum can distinguish the same graphs as the full spectral approach. However, if we remove the spectral information, the expressivity drops significantly — note that SpectRe without spectrum corresponds to RePHINE. Thus, the comparison to RePHINE already represents an ablation study.
>
> Additionally, we have run experiments using LS (Laplacian Spectrum) on BREC datasets. The results are:
>
> | Dataset         | PH⁰  | PH¹  | RePHINE | SpectRe | **LS** |
> |-----------------|------|------|---------|---------|--------|
> | Basic (60)      | 0.03 | 0.97 | 0.97    | 1.00    | **1.00** |
> | Regular (50)    | 0.00 | 0.94 | 0.94    | 1.00    | **1.00** |
> | Extension (100) | 0.07 | 0.70 | 0.77    | 1.00    | **1.00** |
> | CFI (100)       | 0.03 | 0.16 | 0.28    | 0.77    | **0.72** |
> | Distance (20)   | 0.00 | 0.00 | 0.00    | 0.05    | **0.05** |
>
> The results show that LS perform on par with SpectRe on 4 out of 5 BREC datasets. However, the performance on CFI confirms the higher expressivity of SpectRe compared to LS --- 0.77 (SpectRe) vs. 0.72 (LS). Indeed, LS is a simplified version of SpectRe, and is also a contribution of this paper --- as far as we know, no prior work has exploited spectral information persistently in graph learning
>
> Overall, these results demonstrate the importance of spectral information to the expressive power of the proposed descriptors. We will add these experiments to the revised paper
>
> ```
> Q1:How sensitive is SpectRe to noise or perturbations in the filtration functions, particularly when injectivity is violated?
> ```
> Thank you for raising this question. We will quantify an explicit bound on how much SpectRe changes when injectivity is violated, which also may serve as a partial answer to the next question. First of all, we note that perturbing fv does not break stability. This is because the actual filtration SpectRe is being done on is with respect to the edge-level filtration (ex. see Figure 2), and fv is only used to populate with $\alpha$-parameter, which is stable. Thus, SpectRe is globally stable in f_v and locally stable in $f_e$
>
> For simplicity, let G be a graph and let $f_e$ be an injective edge filtration function with values $a1 < ... < an$ and $ai = f(ei)$ for ei an edge color type. Let $g_e$ be a perturbation of $f_e$ such that $g_e(ej) = f_e(ej)$ for all j != i, and $g_e(e_{i}) = g_e(e_{i+1}) = a_{i+1}$ - in other words, $g_e$ merges the i-th and i+1-th value together. Since SpectRe is globally stable in vertex filtration function, we might as well set $f_v = g_v$. We seek to quantify a bound on the distance
>
> $D := d^{Spec R, 0}_B(SpectRe(G, fv, fe), SpectRe(G, gv, fe))$
>
> (The case for 1-dim components is similar). Now observe that the SpectRe diagram for both are the same outside of the vertex deaths at time $a_i, a_{i+1}$ for $f_e$ and time $a_{i+1}$ for $g_e$, so they can be cancelled out under a suitable bijection. Furthermore, the vertex deaths at time $a_{i+1}$ for $f_e$ are a subset of vertex deaths at time $a_{i+1}$ for $g_e$, such that they have the same \alpha and \rho parameters, so we extend the bijections to match them too. Finally, we match the vertex deaths at time $a_{i}$ for $f_e$ to the remaining ones left in $a_{i+1}$ for $g_e$. This bijection gives the bound
>
> $D \leq (a_{i+1} - a_i) + (f_e - g_e)_{\infty} + $
>
> $ \max_{(H1, H2) \in Y} d^{Spec}(p(H1), p(H2)) $
>
> $ = 2(a_{i+1} - a_i) + \max_{(H1, H2) \in Y} d^{Spec}(p(H1), p(H2)) $
>
> where Y is the collection of pairs (H1, H2) where H1 is a connected graph a vertex v died at time a_i for f_e is in, H2 is a connected graph the same vertex v died at time a_{i+1} for g_e is in, and H1 is in H2. p(H1) is the list of non-zero Laplacian eigenvalues of H1.
>
> By the interlacing theorem (see Prop. 3.2.1 of [1]), $d^{Spec}(p(H1), p(H2))$ can be written as $tr(L(H2)) - tr(L(H1))$, where tr is the trace and L(-) is the graph Laplacian. The trace of Laplacian is also the sum of degrees of the graph, which is also twice the number of edges. Thus, we have that
>
> $D \leq 2(a_{i+1} - a_i) + 2 \max_{(H1, H2) \in Y} |E(H_2)| - |E(H_1)|.$
>
> Here we proved a bound of a perturbation where we "pushed a_i up to a_{i+1}", a similar bound can be obtained where we ``push a_{i+1} down to a_i". Also note that Y can be checked component-wise rather than vertex-wise as a simplification.
>
> This is a very interesting question, and we will include this discussion in the main text and the appendix.
> ```
> Q2:Can the stability results be extended to include non-injective filtrations in some approximate sense?
> ```
> Thank you for this question. As in the last answer, it is totally okay for f_v to not be injective. Please also see the discussion above for a quantitative bound on the extent SpectRe fails to be stable for non-injective filtrations.
>
> Here, we also note that the proof of Theorem 4.5 essentially shows that the following local stability result is possible on a non-injective filtration f_e. Let c1, ..., ck be the possible values of f_e on edges, and let x_i^1, ..., x_i^{ai} be f_e^{-1}(ci). Let Z be the subspace of edge filtration functions g_e: X x X/~ \to R_{>0} such that $g_e(f_e^{-1}(ci)) = di$ and $g_e^{-1}(di) = f_e^{-1}(ci)$ (for some di). Z is a metric space, and essentially the same arguments for Theorem 4.5 would show that SpectRe is locally stable around $f_e \in Z$.
>
> ```
> Q3:Would using approximate spectral methods (e.g., Nyström) change expressivity, or just reduce fidelity?
> ```
> Thanks for your great question. Assuming (some of) the filtered subgraphs have low rank structure, we can approximate their corresponding Laplacians with the Nystrom Method. Under low rank structure, this Nystrom approximation would not impact expressivity. Then, one can compute the spectrum on the matrix obtained via Nystrom’s approximation.
>
> In fact, we can use this to further speed up FastSpectre --- by estimating the lower spectrum based on the Nystrom's approximation.
>
> We believe that it might also be interesting to explore the use of  approximate “leverage scores” to find nodes that are most relevant at any timestep. This could make the method more interpretable.
>
> ---
> Thank you for the insightful review and constructive feedback. We hope your concerns are satisfactorily addressed, and if so, would appreciate if you could revisit your score to reflect the same. We are also committed to engage further if you have any more questions or suggestions.
>
> [1] Spectra of graphs. Brouwer et al
> [2] Curvature Filtrations for Graph Generative Model Evaluation. Southern, J. Wayland, M. Bronstein, B. Rieck

---

> > ### Comment · Reviewer_77B6 · 2025-08-06
> > **Response**
> >
> > Thank you for the rebuttal. I appreciate the numbers for FastSpectRe; these are helpful and demonstrate that the method can be scaled to larger datasets in practice. On the empirical side, I still think the downstream improvements in Table 3 are modest and in many cases not clearly statistically significant. However, the additional experiments in the appendix (e.g., Graph Transformers, LS on BREC) help clarify the relative contributions of each component and this goes part of the way toward addressing my request for ablations. The expanded discussion on stability and the injectivity assumption was useful and I appreciate the formal attempt to quantify the impact of injectivity violations; including this in the main text would strengthen the paper.
> >
> > Overall, the rebuttal addressed several concerns I had, particularly around scalability and clarity of the theoretical assumptions. While I still have some reservations about the practical impact of the method on real-world graph learning tasks, I now lean towards a weak accept.

---

### Official Review · Reviewer_jmrk · 2025-07-05

**Clarity:** 2
**Significance:** 2
**Originality:** 2
**Rating:** 4
**Confidence:** 2

**Summary:**

The paper extends the existing work in the line of using Persistent Homology (PH) methods for Graph Representation Learning.

More specifically, the authors extend the existing work of Immonen et. al. [28] by incorporating Laplacian spectral information into the PH invariants, which are then fed into the filtration analysis typical of PH-methods for GNNs.

**Questions:**

"189 The intuition behind why RePHINE cannot differentiate the
190 examples in Figure 3 is that it has the same amount of expres191 sive power as counting the number of connected components
192 and independent cycles (i.e. the harmonic components of the
193 Laplacian) on a monochromatic graph .. "

Please cite this as a theorem from an earlier paper, or prove it somewhere.

**Ethical Concerns:**

["NO or VERY MINOR ethics concerns only"]

**Final Justification:**

The authors have addressed my main concerns via detailed feedback (SOTA GNNs, Spectral Information, Stability). Overall, I am not convinced if the paper will have a strong impact on the field: This is why I have updated my score to a borderline accept.

**Limitations:**

yes

**Paper Formatting Concerns:**

-

**Quality:**

2

**Strengths And Weaknesses:**

Strengths
1. The exposition is very clear.
2. The stability analysis is quite thorough.

Weakness

1. The experimental set-up is mainly designed to compare the results of this paper vs the Immomen paper. There are no comparisons with other state-of-the-art GNNs. In particular, I would like to see a baseline of GNNs which use spectral information in a simple manner (there are tons of models, message-passing based or transformed based which do this). The theoretical results (Theorem 3.2, Theorem 3.3) do not seem very hard to prove.

Overall, the paper seems to be focused on exceeding the results in the Immomen paper. Perhaps PH enthusiasts can better describe the significance of these results (stability section): The broader significance of these results is not so clear to me.

2. I do not see how the stability section is connected with the rest of the paper. Why are there no experimental results to demonstrate the significance of these results?

3. The computational costs of SpectRe are very prohibitive, compared to SOTA GNNs.

---

> ### Author Rebuttal · Authors · 2025-07-31
>
> Thanks for your review. We reply to your comments below.
>
> ```
> There are no comparisons with other state-of-the-art GNNs.
> ```
> Thanks for your comment. We provide additional results in **Table 6 in the Appendix**, where we show that our topological descriptor can boost the performance of Graph Transformers (SOTA GNN). Importantly, since we propose a new persistent topological descriptor that can be readily integrated into graph models, we focused on comparing it to existing topological descriptors. In this regard, REPHINE and the standard PH procedure are the main baselines.
> ```
> In particular, I would like to see a baseline of GNNs which use spectral information in a simple manner.
> ```
> A common approach to incorporating spectral information in graph models is through Laplacian Positional Encoding (LapPE), as used by the Graph Transformer in Table 6. Notably, our results demonstrate that SpectRe can further enhance the performance of models that already leverage spectral features, highlighting its complementary benefits.
>
> To assess the impact of a simpler spectrum-based descriptor on expressivity, we have run experiments using LS (Laplacian Spectrum) on BREC:
>
> | Dataset         | PH⁰  | PH¹  | RePHINE | SpectRe | **LS** |
> |-----------------|------|------|---------|---------|--------|
> | Basic (60)      | 0.03 | 0.97 | 0.97    | 1.00    | **1.00** |
> | Regular (50)    | 0.00 | 0.94 | 0.94    | 1.00    | **1.00** |
> | Extension (100) | 0.07 | 0.70 | 0.77    | 1.00    | **1.00** |
> | CFI (100)       | 0.03 | 0.16 | 0.28    | 0.77    | **0.72** |
> | Distance (20)   | 0.00 | 0.00 | 0.00    | 0.05    | **0.05** |
>
> The results show that LS performs on par with SpectRe on 4/5 datasets. The performance on CFI confirms the higher expressivity of SpectRe compared to LS --- 0.77 vs. 0.72. Indeed, LS is a simplified version of SpectRe, and is also a contribution of this paper --- as far as we know, no prior work has tried to incorporate spectral information persistently in graph learning.
>
> ```
> The theoretical results (Theorem 3.2, Theorem 3.3) do not seem very hard to prove.
> ```
> Thank you for the opportunity to clarify our work. We want to emphasize that our main theoretical results are not just Theorem 3.2,3.3 but also **Theorem 3.4, Theorem 4.4, and Theorem 4.5**. We agree that the results of Theorem 3.2,3.3 are not surprising - it is included for book-keeping purposes.
>
> **Theorem 3.4** shows that augmenting RePHINE with the non-zero eigenvalues of Laplacians is just as expressive as augmenting with the entire family of persistent Laplacian spectra in [1]. This was quite surprising and non-trivial for us, since the definitions of persistent Laplacian are a lot more involved and produce a much larger list. **Theorem 3.4**, however, shows that there are no additional gains that persistent Laplacians would give in this case in terms of expressivity than the more traditional graph Laplacians. Using essentially the proof of Theorem 3.4 in the appendix, the same proof holds for augmenting any descriptor that is at least as expressive as PH. In other words, for any descriptor X such that $X \succeq PH$, augmenting X with the non-zero eigenvalues of graph Laplacians has the same expressive power as adding the entire family of persistent Laplacian spectra.
>
> Please see the answer to the comment below for why proving stability (Section 4) was also quite involved.
>
> ```
> The broader significance of these results [stability] is not so clear to me.
> ```
> Thank you for raising this comment. We want stability results for both RePHINE and SpectRe because they ensure that small perturbations in the input filtrations will not change the output drastically. This is quite important in graph representation learning because often times the attributes/colors we put on the vertices of the graph may be subject to slight measurement errors or imprecision, and the corresponding output of the topological descriptors (TDs) may be affected by that as well. If the TDs are not stable, then their outputs would vary heavily, rendering the method not robust / reliable. This is why the (global) stability of RePHINE and the (local) stability of SpectRe in the paper are quite important, as they ensure the methods are sufficiently robust to perturbations.
>
> From a theoretical viewpoint, there were some difficulties we overcame in proving stability for RePHINE and SpectRe that may be quite helpful for proving stability of persistent descriptors whose tuple size is greater than 2 in the future. For RePHINE, the original proof for Bottleneck stability is not applicable to our setting because it does not account for the $\alpha$ and $\gamma$ parameters. In the appendix, to prove **Theorem 4.4**, we had to come up with the concept of a ``vertex-edge pairing" in **Definition A.3**. Going from RePHINE to SpectRe, we even lose the global stability of PH and RePHINE as seen in Example 4.6. In order to quantify a weaker notion of stability, we used configuration spaces (ie. Conf_n(R)) to describe a local version of stability to correctly describe the stability of SpectRe. We also want to note that the stability of RePHINE was an open question in the original RePHINE paper.
>
> ```
> Why are there no experimental results to demonstrate the significance of these results [stability]?
> ```
> To address your issue, we look at the first 4 graphs from BREC (basic.npy). For each graph, we define ($f_v$, $f_e$) with $f_v$ being the degree of the vertex and $f_e$ being the average of the degree of the two vertices. The following table shows the bottleneck distance and the inequality bound $3||f_e - g_e|| + ||f_v - g_v||$ for different choices of $(g_v, g_e)$.
>
> | $g_v$              | $g_e$              | **Graph ID** | **RePHINE Dist** | **RePHINE Bound** | **SpectRe Dist** | **SpectRe Bound** |
> |--------------------|--------------------|--------------|------------------|-------------------|------------------|-------------------|
> | $\sin(10 f_v)$     | $5 f_e$            | G1           | 65.25            | 78.30             | 66.23            | 78.30             |
> |                    |                    | G2           | 55.25            | 66.30             | 56.30            | 66.30             |
> |                    |                    | G3           | 68.25            | 86.99             | 69.25            | 86.99             |
> |                    |                    | G4           | 62.26            | 72.30             | 62.26            | 72.30             |
> | $0.1f_v+0.1$       | $\exp(-f_e)$       | G1           | 21.18            | 24.19             | 71.95            | 24.19             |
> |                    |                    | G2           | 17.77            | 20.28             | 65.98            | 20.28             |
> |                    |                    | G3           | 22.59            | 26.60             | 72.37            | 26.60             |
> |                    |                    | G4           | 18.78            | 21.79             | 60.60            | 21.79             |
>
> The outputs of the table are expected behaviors. We first see that no matter what the $(g_v, g_e)$ is, the RePHINE distance is always bounded by the RePHINE bound, which corresponds to RePHINE being **globally stable**.
>
> For SpectRe, when $g_v = \sin(10 f_v), g_e = 5 f_e$, we observe that we did not break stability. This is because of two reasons (1) changes in the vertex-filtration function do not affect stability, and (2) the change $f_e\to 5g_e$ did not cross the region of non-injectivity. When $g_v = 0.1f_v+0.1, g_e=\exp(-f_e)$, we see the stability of SpectRe is broken. This is expected behavior because $\exp(-f_e)$ is an order reversing function, and the path from $f_e$ to $\exp(-f_e)$ would necessarily cross some region of non-injectivity.
>
> ```
> The computational costs of SpectRe are very prohibitive, compared to SOTA GNNs.
> ```
> Indeed, the vanilla version of SpectRe is computationally expensive. To make it scalable to larger datasets, we developed FastSpectRe by combining efficient algorithms for computing partial spectra with a scheduling strategy. The average times per epoch (measured over 10 epochs) on CPU for FastSpectre (with scheduling) on the largest datasets are as follows: NCI1 / NCI109 (11s), IMDB-BINARY (10s), ZINC (17s), and MOLHIV (130s).
>
> ```
>  Please cite this as a theorem from an earlier paper, or prove it somewhere.
> ```
> We will prove it here as follows and include in the paper.
>
> **Lemma:** Let G and H be two graphs with the same number of nodes with one single color, then RePHINE can differentiate G and H if and only if either $b_0(G) \neq b_0(H)$ or $b_1(G) \neq b_1(H)$.
>
> Here $b_0, b_1$ refer to the 0th and 1st Betti numbers of G and H, which are the number of connected components and independent cycles.
>
> **Proof:** Since G and H only have one color, $f_v, f_e$ are both constant functions of values $a_V, a_E$, so the $\alpha$ and $\gamma$ parameters of RePHINE for $G$ and $H$ are the same and can be discarded. Here, the filtrations with respect to $f_e$ has two steps - it first spawns all the vertices and then adds in all the edges. Thus, RePHINE can differentiate G and H if and only if looking at its first two parameters alone can differentiate $G$ and $H$. Since $f_v$ and $f_e$ are constant, a vertex either dies at time $a_V$ or at infinity, and all independent cycles are born at time $a_E$. Thus RePHINE can differentiate G and H if and only if they have different counts of pairs (0, a_V), (0, \infty), and (1, a_E). Since G and H have the same number of vertices, these counts differ if and only if b_0(G) != b_0(H) and b_1(G) != b_1(H).
>
> ---
> We hope your concerns have been satisfactorily addressed, and if so, would appreciate if you could revisit your score to reflect the same. We are also committed to engaging further if you have any additional questions, concerns, or suggestions.

---

> > ### Comment · Reviewer_jmrk · 2025-08-06
> > **Rebuttal Response**
> >
> > Thanks to the authors for addressing my concerns and providing thorough arguments. I will increase my score to reflect the detailed feedback.

---

> > > ### Author Response · Authors · 2025-08-07
> > >
> > > Thank you for your positive comment! We are glad that your concerns have been resolved, and we will incorporate your helpful feedbacks in the revision.

---

### Official Review · Reviewer_aJUR · 2025-07-06

**Clarity:** 3
**Significance:** 2
**Originality:** 2
**Rating:** 4
**Confidence:** 4

**Summary:**

This paper introduces SpectRe, a topological descriptor for graphs that enhances persistent homology (PH) diagrams with spectral information. Designed to be more expressive than prior methods like RePHINE, SpectRe augments persistence tuples with eigenvalues of the graph Laplacian at critical filtration steps. The authors provide a theoretical analysis of the descriptor's expressivity and local stability, and demonstrate its effectiveness on synthetic isomorphism tests and real-world graph learning tasks.

**Questions:**

1. **On the Choice of Persistent Laplacian Construction:** The paper argues that persistent Laplacians offer no more expressive power than the standard graph Laplacian. However, the cited work [49] is one of several constructions. How does this result hold up when considering other definitions, such as the one proposed by Mémoli, Wan, and Wang in "Persistent Laplacians: properties, algorithms and implications"?
2. **On Comparison with Multi-Parameter Persistence:** The paper builds upon single-parameter persistence. How do you see SpectRe's expressivity in relation to representations derived from Multi-Parameter Persistence (MPH) [1, 2, 3]? Since MPH can capture more complex interactions by using multiple simultaneous filtrations, could replacing the single-parameter PH backbone of SpectRe with an MPH backbone lead to an even more powerful descriptor?
3. **On Local vs. Global Stability:** Could you provide more intuition on the practical implications of local stability? Are there realistic scenarios or types of graph data where you would expect the lack of global stability to be a significant issue?
4. **On the Choice of Filtration:** The performance of SpectRe can be dependent on the choice of the initial vertex and edge filtration functions. Do you have any general guidelines or heuristics for choosing effective filtration functions for a given graph learning task?


**References**

[1] Magnus Bakke Botnan and Michael Lesnick. An introduction to multiparameter persistence. In Representations of algebras and related structures, EMS Ser. Congr. Rep., pages 77–150. EMS Press, Berlin, [2023] ©2023.

[2] Oliver Vipond. Multiparameter persistence landscapes. Journal of Machine Learning Research, 21(61):1–38, 2020.

[3] Xin, C., Mukherjee, S., Samaga, S. N., and Dey, T. K. GRIL: a 2-parameter persistence based vectorization for machine learning. In Proceedings of 2nd Annual Workshop on Topology, Algebra, and Geometry in Machine Learning (TAG-ML), volume 221 of Proceedings of Machine Learning Research, pp. 313–333. PMLR, 7 2023.

**Ethical Concerns:**

["NO or VERY MINOR ethics concerns only"]

**Final Justification:**

I thank the authors for a thoughtful and highly productive discussion. Their rebuttal was exemplary.

My initial concerns centered on the novelty of the paper's main theoretical claims. After a constructive exchange, the authors have agreed to significantly revise the paper to address these points. Specifically:

- On Expressivity (Thm. 3.4): The authors have acknowledged that their claim about the persistent Laplacian's expressivity is limited to the specific case of edge-based graph filtrations, where the result is a direct consequence of known properties established in prior work. They have committed to citing these works and clarifying this important context. Their new counter-example for vertex-based filtrations is an excellent addition that adds valuable nuance and strengthens their discussion.

- On Proof Techniques (Thm. 4.4): I am satisfied with their commitment to properly cite the standard TDA techniques used in their stability proof.

This discussion has clarified my final assessment. The paper introduces SpectRe, a useful descriptor. However, the authors' revisions will now accurately frame the work as a valuable exploration of a specific, practical case (edge-filtrations), with theoretical results that follow from known properties and standard techniques. This limits the overall novelty of the contribution.

For these reasons, I am maintaining my score of 4 (Borderline Accept). The paper is technically solid and the authors' response has been outstanding, but its contribution is ultimately more incremental than groundbreaking.

**Limitations:**

yes

**Quality:**

3

**Strengths And Weaknesses:**

**Strengths:**
- The paper is theoretically sound and provides a comprehensive analysis of the proposed SpectRe descriptor. The expressivity proofs and the stability property of are well-executed.
- The isomorphism tests on synthetic datasets are well-designed and clearly support the theoretical claims about the superior expressive power of SpectRe.
- The paper is well-written and presents its complex ideas in a structured and understandable manner.

**Weaknesses:**
- The core idea is to add spectral features to an existing descriptor (RePHINE). This is a natural follow-up step rather than a fundamentally new approach. The method essentially decorates an existing descriptor with a well-known useful feature type. The main theoretical result—that adding more information (spectral features) increases the model's ability to distinguish non-isomorphic graphs—is not surprising.
- While theoretically more expressive, the empirical improvements on several real-world datasets (Table 3) are marginal. This raises questions about the practical utility of the proposed method, especially when considering its significant computational overhead.

---

> ### Author Rebuttal · Authors · 2025-07-31
>
> We are grateful to Reviewer aJUR for their time and detailed comments. We reply to your comments/questions below.
>
> ```
> The core idea is to add spectral features to an existing descriptor (RePHINE). This is a natural follow-up step rather than a fundamentally new approach.
> ```
> Thank you for giving us the opportunity to clarify our work. Although we chose RePHINE as the base descriptor to incorporate spectral information, we could have created a similar construction with any persistent based descriptors in the given filtration. We chose RePHINE as our base because it is a simple topological descriptor that is provably more expressive than traditional PH. Otherwise, the Laplacian spectrum (LS) (defined after Def 3.1) could have been a more generic option, but its expressivity is bounded by SpectRe.
>
> For Section 3, **Theorem 3.4** shows that augmenting RePHINE with the non-zero eigenvalues of graph Laplacians is just as expressive as augmenting with the entire family of persistent Laplacian spectra in [1]. This was quite surprising for us, since the definitions of persistent Laplacian are a lot more involved and produces a much larger list. **Theorem 3.4**, however, shows that there are no additional gains persistent Laplacians would give in this case in terms of expressivity than the more traditional graph Laplacians. Using essentially the proof of Theorem 3.4 in the appendix, the same proof holds for augmenting any descriptor that is at least as expressive as PH. In other words, for any descriptor X such that $X \succeq PH$, augmenting X with the non-zero eigenvalues of graph Laplacians has the same expressive power as adding the entire family of persistent Laplacian spectra.
>
> In Section 4, there were some difficulties we overcame in proving stability for RePHINE and SpectRe that may be quite helpful for proving stability of persistent descriptors whose tuple size is greater than 2 in the future. For RePHINE, the original proof for Bottleneck stability is not applicable to our setting because it does not account for the $\alpha$ and $\gamma$ parameters. In the appendix, to prove **Theorem 4.4**, we had to come up with the concept of a ``vertex-edge pairing" in **Definition A.3**. Going from RePHINE to SpectRe, we even lose the global stability of PH and RePHINE as seen in Example 4.6. This was quite surprising to us during discovery. In order to quantify a weaker notion of stability, we used configuration spaces (ie. Conf_n(R)) to describe a local version of stability to correctly describe the stability of SpectRe.
>
> We also want to note that the ``vertex-edge pairing" in Def A.3 and the proofs can give approaches to proving the stability of persistent descriptors whose tuple size is greater than 2.
>
> ```
> The main theoretical result ... is not surprising.
> ```
> Thank you for the opportunity to clarify our work. We want to emphasize that our main theoretical results are not just Theorem 3.3 (on SpectRe being more expressive) but also **Theorem 3.4, Theorem 4.4, and Theorem 4.5**. We agree that the results of Theorem 3.3 are not surprising - it is included for book-keeping purposes. Please see our comment to the previous question for we find these results to be surprising.
>
> ```
> While theoretically more expressive, the empirical improvements on several real-world datasets (Table 3) are marginal. This raises questions about the practical utility of the proposed method, especially when considering its significant computational overhead.
> ```
> We note that the empirical gains over RePHINE are consistent across datasets. In addition, our topological descriptors can be readily integrated into SOTA graph models, and we have reported additional evidence of their benefits in **Table 6 (Appendix)** through the combination of **SpectRe and Graph Transformers**. Using topological descriptors boosts the performance of the graph Transformer in 3 out of 4 datasets. Also, the gains achieved by SpectRe are higher than those of RePHINE.
>
> Most important, our expressivity results are significant and clearly demonstrates benefits of our proposal over existing topological descriptors, which is the primary motivation for our proposal.
>
> ```
> On the Choice of Persistent Laplacian Construction: The paper argues that persistent Laplacians offer no more expressive power than the standard graph Laplacian. However, the cited work [49] is one of several constructions. How does this result hold up when considering other definitions, such as the one proposed by Mémoli, Wan, and Wang in "Persistent Laplacians: properties, algorithms and implications"?
> ```
> Thank you for the question. Actually, the definition proposed by Mémoli, Wan, and Wang in "Persistent Laplacians: properties, algorithms and implications" is equivalent to the cited work's definition in [1]. At the beginning of Section 2.2 in the arXiv v3 of the paper, when the authors Mémoli, Wan, and Wang defined the persistent Laplacian citing precisely the paper in [1]. The authors also directly attributed the definition to come from [1] (and indepdently in [2]) on page 3 of the introduction. We are also not aware of an alternative construction for the persistent combinatorial Laplacians.
>
> ```
> On Comparison with Multi-Parameter Persistence
> ```
> Thank you for the question. We looked more into multi-parameter persistence in the context of our setup, and we have found an example where 2-parameter PH can give more expressive power descriptors than PH alone. SpectRe can be readily adapted to this 2-parameter scheme as well, and we are confident it can make a more powerful descriptor, although we have not investigated an example yet.
>
> In multi-parameter persistence, we define a function f: G -> R^n. There is no time anymore, but rather a partial order on R^n. In our case, we can define a 2-parameter filtration as (fv, fe): G -> R^2, with G_{s,t} = (f_v, f_e)^{-1}( (-infty, s] x (-infty, t]) for (s,t) in R^2.
>
> In this case, we found a pair of graphs that PH for fv and fe (ie. Def 2.2) cannot tell apart, but the 2-parameter scheme above can. Let G (left) and H (right) be the graphs
> ```
> R - R - B     B - R - R - B
>      \---B
> ```
> In this case, one can check PH in Def 2.2 cannot tell them apart for any choices of fv and fe. Now let fe(R-B) = 1, fe(R-R) = 2, fv(B) = 1, fv(R) = 2. Consider the sub-filtration of the 2-parameter filtration given by G_{*, 1}, ie. (G_{0, 1} -> G_{1, 1} -> G_{2, 1}.) One can check that this is the vertex-coloring filtration fv on the subgraphs:
> ```
> R   R - B    B - R  R - B
>      \---B
> ```
> In this case, the left one gives persistence pairs (1, 2), (2, 2), (2, infty), (1, infty) (since a R and a B die here at t=2), but the right one gives pairs (1, infty), (1, infty), (2, 2), (2,2) (since two R's die here at t=2). We will include this as an interesting future direction in the paper.
>
> ```
> On Local vs. Global Stability: Could you provide more intuition on the practical implications of local stability? Are there realistic scenarios or types of graph data where you would expect the lack of global stability to be a significant issue?
> ```
> Thank you for your question. The idea for **global vs. local stability** is as follows. Recall $f_v$ is a  function $f_v: X \to \mathbb{R}$ and $f_e$ is a function $f_e: X x X/~ \to \mathbb{R}_{>0}$. Since the domains of $f_v, f_e$ are both finite with say size a and b, we can identify $f_v$ as a point in $\mathbb{R}^a$ and $f_e$ as a point in $\mathbb{R}^b$ (after we fix an ordering on the domains). Every point of $\mathbb{R}^{a+b}$ can thus be viewed as a pair $(f_v, f_e)$.
>
> Intuitively, RePHINE is **globally stable** in the sense that it is a continuous on $\mathbb{R}^{a+b}$ with respect to the bottleneck distance in the codomain and a suitable metric on $\mathbb{R}^{a+b}$ given by the bound in **Theorem 4.4**.
>
> Intuitively, SpectRe is **locally stable** in the sense that it is only continuous on $\mathbb{R}^{a+b}$ with a subset (the region of non-injectivity in the paper) removed, which typically has multiple connected components. When SpectRe crosses from one component to other, the output jumps drastically and is not continuous. It is only locally continuous.
>
> Numerically, what this means is - for **global stability**, any two pairs of filtration functions are bounded by the inequality in the paper. For **local stability**, only a pair X of injective filtraiton functions and another pair sufficiently close to X satisfies the same inequality.
>
> We do not expect the lack of global stability to be a significant issue, since we want stability to ensure that tiny perturbations in the input filtrations do not change the output drastically. For sufficiently small perturbations, the local stability would always suffice.
>
> ```
> On the Choice of Filtration: The performance of SpectRe can be dependent on the choice of the initial vertex and edge filtration functions. Do you have any general guidelines or heuristics for choosing effective filtration functions for a given graph learning task?
> ```
> Thank you for your question. Both fixed and learnable filtration functions have been explored in the literature. Fixed functions offer advantages in terms of computational efficiency and interpretability. In contrast, recent works (e.g., Topological GNNs, ICLR 2022) have applied learnable filtration functions, which adapt to the task and data, potentially leading to improved performance. In our real-world experiments, we opted for learnable filtration functions to allow greater flexibility and task-specific adaptation.
>
> ---
>
> Many thanks for your insightful review and constructive feedback. We hope your concerns have been satisfactorily addressed, and if so, would appreciate if you could revisit your score to reflect the same. We are also committed to engaging further if you have any additional questions, concerns, or suggestions.
>
> [1] Persistent spectral graph. R. Wang, D. Nguyen, G. Wei.
> [2] Talk: Persistent harmonic forms. A. Lieutier.

---

> > ### Comment · Reviewer_aJUR · 2025-08-05
> >
> > Thank you for your detailed response to my review, particularly the discussion on multi-parameter persistence.
> >
> > I believe some of my main concerns regarding the theoretical conclusion on **the expressivity of persistent Laplacian** warrants further discussion.
> >
> > I understand that the formal operator definitions for the persistent Laplacian are basically equivalent across the cited works. My question, however, is aimed at the practical construction and application of the persistent Laplacian within your theoretical framework. The proof of your equivalence claim in Theorem 3.4 appears to rest on an analysis of filtrations that are restricted to 1-dimensional complexes (graphs). In this specific setting, the persistent 1-Laplacian is simplified because the absence of 2-simplices trivializes the $∂_2$ boundary map. This essentially means the comparison is being made against a version of the persistent Laplacian that cannot realize its full potential. The true power of the persistent laplacian operator lies in its application to higher-dimensional simplicial filtrations where the interplay between 1-cycles and 2-simplices is non-trivial.
> >
> > Because the analysis is confined to a setting that neutralizes the core mechanism of the high-order persistent Laplacian, the conclusion that it offers no more expressive power than SpectRe feels specific to this oversimplified construction. It would strengthen the paper to include a discussion of this limitation, clarifying that the claim holds for graph-only filtrations but may not generalize to richer simplicial filtrations (e.g. rips complex) where the full power of the persistent Laplacian would be unleashed.

---

> > > ### Author Response · Authors · 2025-08-05
> > >
> > > Thank you for your continued engagement! We reply to your comments below.
> > > ```
> > > The proof of .. Theorem 3.4 appears to rest on an analysis ... 1-dim complexes (graphs). ... It would strengthen the paper to include a discussion of this limitation, clarifying that the claim holds for graph-only filtrations but may not generalize to richer simplicial filtrations (e.g. rips complex) where the full power of the persistent Laplacian would be unleashed.
> > > ```
> > > Thank you for your comment. We agree that our claim in Theorem 3.4 holds because we are working with GNNs and considering graph-based filtrations. If we are working with simplicial complex neural networks (SCNNs), there is no reason to expect that a similar result would hold for SCNNs, as the arguments rely on the 1-dimensionality of the graph.
> > >
> > > There are approaches to GNNs, such as the Vietoris-Rips (VR) filtration you mentioned, that builds a simplicial complex from the graph, and then consider filtrations on the new simplicial complex. This was not the focus of our paper, as we only investigated filtrations on the original graph itself. As we will explain below, **VR-filtration based methods**, and many simplicial filtrations in the context of GNNs, may not be directly comparable to the **graph-filtration based methods** we are considering. Thus, the two are really complementary.
> > >
> > > Consider the VR-filtration outlined right under Theorem 1 of [1] - that is, given a graph G=(V,E), we consider the filtration f_V of a simplicial complex K, where K is the set of all non-empty subsets S of V such that the diameter of S is not infinity (ie. they are all reachable from one another), and the filtration is $f_V(S) = max_{u, v} d_G(u, v)$ ($d_G$ is the shortest-path distance)
> > >
> > > Let **VRPH** be the PH of this filtration. We claim that **VRPH** is incomparable to **PH** (Def 2.2) in terms of expressive power. Indeed, let G be a path-graph on 3 vertices and H be a cycle graph on 3 vertices, then **VRPH** cannot differ them (see the discussions right before Section 5 of [1]), but **PH** can due to the presence of a cycle in H that it can detect. The high-level philosophy for why they are incomparable is that the **PH** in our paper is only adding <= 1-dim simplicies, whereas **VRPH** adds in the higher-dimensional simplicies simultaneously that might remove non-trivial information in low-dimensions. (Intuition: a circle has non-trivial 1st Betti number, but if you fill in the circle to a disk, its 1st Betti number is then 0). While adding in higher-dim simplicies can add more information, it also removes some low-dim information. Hence, the two methods are incomparable. This is also the high-level reason behind why **VRPH** is in fact incomparable with WL tests as well (see [1] for more details).
> > >
> > > Thus, rather than viewing VR filtrations as a strictly richer simplicial filtration, the graph filtration we considered and the VR filtrations are really complementary to each other. Note also that the same holds for many general simplicial filtrations that build on the base graph, as adding higher simplicies intermediately may still remove low-dim information. There is no reason to expect the two to be comparable if we also add in spectral information (ex. the following question may likely be incomparable - VR-filtration + persistent Laplacian vs. the Laplacian Spectrum (LS) or SpectRe in our paper).
> > >
> > > A question that likely has a T/F answer would be - is VR + persistent Laplacian > VR + combinatorial Laplacians? We believe the answer is likely yes.
> > >
> > > We do note that, if a simplicial filtration adds in the entire graph first, and then adds in the higher dimensional simplicies at a later time, then it does subsume the context of a graph-based filtration. However, a full simplicial filtration may in general be quite difficult to enumerate when the scale of the base graph gets larger. Ex. For a connected graph G with n vertices, the full VR filtration on G will go through 2^n - 1 many simplicies. This is exponential in scale. SpectRe, in contrast, while incomparable in terms of expressivity to VR-based methods, is polynomial.
> > >
> > > Thank you again for raising this point. We will for sure add a discussion on (1) how Theorem 3.4 is limited to graph-based filtrations in the context of GNNs, (2) how some simplicial filtrations (including VR) in the context of GNNs are complementary to our set-up of graph-based filtrations rather than subsuming each other, (3) some interesting future directions for persistent Laplacians vs. combinatorial Laplacians on simplicial filtrations for GNNs, and (4) how SpectRe still keeps a polynomial computational costs as opposed to the possible exponential costs a simplicial filtration method.
> > >
> > > ---
> > > We hope this addresses your concerns, and, if so, would appreciate if you could change your rating to reflect the same.
> > >
> > > [1] On the Expressivity of Persistent Homology in Graph Learning. Rubén Ballester and Bastian Rieck. Learning on Graphs Conference (LoG) 2024

---

> > > > ### Comment · Reviewer_aJUR · 2025-08-08
> > > >
> > > > Thank you for your detailed response. While I appreciate the planned revisions, my primary concerns regarding the novelty and contextualization of the paper's theoretical contributions remain.
> > > >
> > > > **1. On the Novelty of Theorem 3.4:**
> > > >
> > > > The core claim of Theorem 3.4 is that on graph-only filtrations, the persistent Laplacian spectrum offers no more expressivity than the standard graph Laplacian spectrum. This result appears to be a direct consequence of previously established properties rather than a novel finding. Specifically:
> > > >
> > > > The simplification of the persistent laplacian to the graph laplacian for graph edge filtrations have been described as a basic or trivial case in prior foundational work (e.g.[1,2]). Given that, the conclusion of the theorem appears to be an aggregation of existing results. The paper should properly cite and discuss this prior work to accurately position its contribution.
> > > >
> > > > **2. On Citing Standard Proof Techniques for Theorem 4.4:**
> > > >
> > > > Furthermore, regarding the stability proof for Theorem 4.4, the "vertex-edge pairing" technique is a common and powerful method in computational topology. It is well-known and used in many proofs and algorithms in the TDA field (e.g., Ch. 3 & 10 in [3]  ). The manuscript should properly cite and acknowledge this standard machinery.
> > > >
> > > > **In Summary:**
> > > >
> > > > While the SpectRe descriptor is a useful construction, the paper's main theoretical claims appear to overstate their novelty by not fully contextualizing them within established TDA literature. The promised revisions are a good start, but for the paper to be a strong contribution, it must also thoroughly address these fundamental issues of citation and novelty for both the main theorems and their proofs.
> > > >
> > > >
> > > > **Reference:**
> > > >
> > > > [1] The Persistent Laplacian for Data Science: Evaluating Higher-Order Persistent Spectral Representations of Data, Thomas Davies, Zhengchao Wan, Ruben J Sanchez-Garcia Proceedings of the 40th International Conference on Machine Learning, PMLR 202:7249-7263, 2023.
> > > >
> > > > [2] Persistent Laplacians: Properties, Algorithms and Implications, Facundo Mémoli, Zhengchao Wan, Yusu Wang, SIAM Journal on Mathematics of Data Science, 4(2):858–884, 2022
> > > >
> > > > [3] Computational Topology for Data Analysis, Tamal K. Dey, Yusu Wang. Cambridge U. Press.

---

> > > > > ### Author Response · Authors · 2025-08-09
> > > > >
> > > > > Thank you for your continued engagement and for pointing us to these references. We will be sure to appropriately credit the works you mentioned to better position the contributions of our work. We address your comments below.
> > > > >
> > > > > ```
> > > > > Novelty of Theorem 3.4
> > > > > ```
> > > > > Thank you for your constructive feedback. We agree that the 0-dim persistent Laplacian part of Theorem 3.4 has appeared in equivalent forms in [1,2]. Although neither reference explicitly discussed why 1-dim persistent Laplacians can be recovered by graph Laplacian + PH, they directly reduce to 1-dim Laplacians for graphs, and standard linear algebra shows that they have the same non-zero eigenvalues as the graph Laplacian. The zero eigenvalues cannot be recovered by graph Laplacians alone, but has multiplicity as the 1st homology.
> > > > >
> > > > > In our revision, we will be sure to discuss this and acknowledge the significance of [1,2], and direct to some filtrations other than edge-filtrations where the two may differ, including the Vietoris-Rips filtration you mentioned earlier.
> > > > >
> > > > > To add an interesting example to the discussion, the proof of Theorem 3.4 follows from Corollary B.6 - that the non-zero eigenvalues of persistent Laplacians in edge-based filtrations on graphs can be recovered by the eigenvalues of the graph Laplacians. We note this is however not true for vertex-based filtrations.
> > > > >
> > > > > Consider a two step filtration $K \subset L$ where $L$ is the path-graph on 4-vertices labeled 1-2-3-4 and $K$ is the discrete subgraph {1,4}. This filtration is the vertex-based filtration of a function $f_v$ given by $f_v(1) = f_v(4) = 1$ and $f_v(2) = f_v(3) = 2$. If we only look at the graph Laplacian spectra of the filtration, we would get that $K$ has eigenvalues 0,0 and $L$ has eigenvalues $0,2-\sqrt{2},2,2+\sqrt{2}$.
> > > > >
> > > > > The persistent 0-dim Laplacian of the pair $(K, L)$ is the matrix [[1/3,-1/3],[-1/3,1/3]] with eigenvalues $0, 2/3$. This can be verified using the Matlab code in [2] with inputs
> > > > > ```
> > > > > B1 = [0 0];
> > > > > B2 = [-1 0 0; 1 -1 0; 0 1 -1; 0 0 1];
> > > > > Gind = [1 4];
> > > > > ```
> > > > > The extra 2/3 cannot be recovered from the graph Laplacian spectra of K and L alone. Indeed, for a different filtration of L with K' = {1, 3}, we would get the matrix [[1/2,-1/2],[-1/2,1/2]] with eigenvalues $0, 1$.
> > > > >
> > > > > Based on this, we can possibly enhance the power of SpectRe by adding a 6th parameter to include the spectral info for vertex-filtrations (it makes the TD symmetric too). Thank you for the exciting possibility your feedback opened!
> > > > >
> > > > > ```
> > > > > Theorem 4.4
> > > > > ```
> > > > > Thank you for suggesting the reference in [3]! We will be sure to include it and position appropriately. To clarify, when we said "vertex-edge pair" in our initial comment, we meant the context in which it is used rather than the concept itself (for which we are not surprised has been invented). In Chapter 3 of [3], vertex-edge pairs were mentioned in a remark about connections to minimum spanning trees. In Chapter 10 of [3], vertex-edge pairs were used in algorithmic applications with discrete Morse theory. In our work, we used the pair to estimate and bound the variant of Bottleneck distance we posed on RePHINE and SpectRe. The usage of the pair in [3] was primarily algorithmic rather than in the context of stability.
> > > > >
> > > > > We also note that the principle of vertex-edge pairs was also used in Theorem 4.5, which is on the local stability of SpectRe. During discussion with **Reviewer 77B6**, we also obtained an explicit bound for how bad the stability can fail when injectivity is violated, which we plan to include as **an additional theoretical result**.
> > > > >
> > > > > Let $G$ be a graph and let $f_e$ be an injective edge filtration function with values $a_1 < ... < a_n$, $a_i = f(e_i)$ for $e_i$ an edge color type. Let $g_e$ be a perturbation of $f_e$ such that $g_e(e_j) = f_e(e_j)$ for all j != i, and $g_e(e_{i}) = g_e(e_{i+1}) = a_{i+1}$ (ie. $g_e$ merges the i-th and i+1-th value together). Let $D = d^{Spec R, 0}_B(SpectRe(G, fv, fe), SpectRe(G, gv, fe))$, then we can find a bound as
> > > > >
> > > > > $$D \leq 2(a_{i+1} - a_i) + 2 \max_{(H_1, H_2) \in Y} |E(H_2)| - |E(H_1)|,$$
> > > > >
> > > > > where Y is the collection of pairs $(H_1, H_2)$ where $H_1$ is the connected component a vertex v died at time $a_i$, for $f_e$, is in, $H_2$ is the connected component the same vertex v died at time $a_{i+1}$, for $g_e$, is in, and $H_1$ is in $H_2$. A similar bound can be obtained for $d^{Spec R, 1}$. Please see the discussion with Reviewer 77B6 for more details on the reasoning.
> > > > >
> > > > > ----
> > > > > We are grateful for your thoughtful remarks and suggestions that have helped improve this work. We will be sure to act on all of them. We hope our response reinforces your support for this work.
> > > > >
> > > > > [1] The Persistent Laplacian for Data Science: Evaluating Higher-Order Persistent Spectral Representations of Data, T. Davies, Z. Wan, R. Sanchez-Garcia.
> > > > >
> > > > > [2] Persistent Laplacians: Properties, Algorithms and Implications, F. Mémoli, Z. Wan, Y. Wang.
> > > > >
> > > > > [3] Computational Topology for Data Analysis, T. K. Dey, Y. Wang.

---

> > > > > > ### Comment · Reviewer_aJUR · 2025-08-09
> > > > > >
> > > > > > > *"from Corollary B.6 - that the non-zero eigenvalues of persistent Laplacians in edge-based filtrations on graphs can be recovered by the eigenvalues of the graph Laplacians. We note this is however not true for vertex-based filtrations. ..."*
> > > > > >
> > > > > > Yes, to note that the equivalence between persistent laplacian and graph laplacian requires the same vertex set, which is satisfied by the edge-based filtration, but not guaranteed by a vertex-based filtration, as shown in the example.
> > > > > >
> > > > > > I thank the authors for their thoughtful and highly constructive response. I do not have any further questions. I will reconsider my rating.

---

### Decision · Program_Chairs · 2025-09-17

**Decision:**

Accept (poster)

**Comment:**

The paper introduces SpectRe, a topological graph descriptor that augments persistent homology diagrams with Laplacian spectral information. This leads to greater expressivity than both RePHINE and spectrum-only approaches, while retaining stability guarantees. The main contribution of the paper lies in providing proofs of expressivity and local stability. The method is also empirically validated on isomorphism benchmarks and real-world GNN tasks.

After the authors clarified some doubts during the rebuttal period, the reviewers agreed that the idea is conceptually clean and appreciated the theoretical results and their exposition. At the same time, they raised concerns about the novelty of the work, since it reuses well-known results from TDA that were not properly acknowledged throughout the paper.

Overall, the reviewers have a positive opinion of the paper.